# The *O*-GlcNAc transferase OGT is a conserved and essential regulator of the cellular and organismal response to hypertonic stress

Sarel J. Urso[1,2], Marcella Comly[3], John A. Hanover[3], Todd Lamitina[1,2,4]*

**1** Graduate Program in Cell Biology and Molecular Physiology, University of Pittsburgh School of Medicine, Pittsburgh, PA, United States of America, **2** Department of Cell Biology, University of Pittsburgh School of Medicine, Pittsburgh, PA, United States of America, **3** Laboratory of Cellular and Molecular Biology, National Institute of Diabetes and Digestive and Kidney Diseases, National Institute of Health, Bethesda, MD, United States of America, **4** Division of Child Neurology, Department of Pediatrics, Children's Hospital of Pittsburgh, Pittsburgh, PA, United States of America

* stl52@pitt.edu

**Data Availability Statement:** All relevant data are within the manuscript and its Supporting Information files.

## Abstract

The conserved *O*-GlcNAc transferase OGT *O*-GlcNAcylates serine and threonine residues of intracellular proteins to regulate their function. OGT is required for viability in mammalian cells, but its specific roles in cellular physiology are poorly understood. Here we describe a conserved requirement for OGT in an essential aspect of cell physiology: the hypertonic stress response. Through a forward genetic screen in *Caenorhabditis elegans*, we discovered OGT is acutely required for osmoprotective protein expression and adaptation to hypertonic stress. Gene expression analysis shows that *ogt-1* functions through a post-transcriptional mechanism. Human OGT partially rescues the *C. elegans* phenotypes, suggesting that the osmoregulatory functions of OGT are ancient. Intriguingly, expression of O-GlcNAcylation-deficient forms of human or worm OGT rescue the hypertonic stress response phenotype. However, expression of an OGT protein lacking the tetracopeptide repeat (TPR) domain does not rescue. Our findings are among the first to demonstrate a specific physiological role for OGT at the organismal level and demonstrate that OGT engages in important molecular functions outside of its well described roles in post-translational O-GlcNAcylation of intracellular proteins.

## Author summary

The ability to sense and adapt to changes in the environment is an essential feature of cellular life. Changes in environmental salt and water concentrations can rapidly cause cell volume swelling or shrinkage and, if left unchecked, will lead to cell and organismal death. All organisms have developed similar physiological strategies for maintaining cell volume. However, the molecular mechanisms that control these physiological outputs are not well understood in animals. Using unbiased genetic screening in *C. elegans*, we discovered that a highly conserved enzyme called O-GlcNAc transferase (OGT) is essential for regulating physiological responses to increased environmental solute levels. A human form of OGT

**Funding:** TL received grants R01GM105655 and R01GM135577 from the NIH. The funders had no role in study design, data collection and analysis, decision to publish, or preparation of the manuscript.

**Competing interests:** The authors have declared that no competing interests exist.

can functionally substitute for worm OGT, showing that this role is conserved across evolution. Surprisingly, the only known enzymatic activity of OGT was not required for this role, suggesting this enzyme has important undescribed molecular functions. Our studies reveal a new animal-specific role for OGT in the response to osmotic stress and show that *C. elegans* is an important model for defining the conserved molecular mechanisms that respond to alterations in cell volume.

## Introduction

Cells must adapt to perturbations in extracellular osmolarity to maintain cell volume, membrane tension, and turgor pressure [1]. Hypertonic stress leads to loss of cell volume, increased intracellular ionic strength, and protein dyshomeostasis. Failure to initiate protective mechanisms against these perturbations leads to cell death [2]. Hypertonicity contributes to several pathophysiological conditions (diabetes) and is also a feature of normal physiological states such as those that exist in the kidney and thymus, which facilitate urinary concentration and T-cell development respectively [3, 4]. Cells in these tissues can survive in hypertonic conditions because of evolutionarily conserved adaptive mechanisms.

Cells adapt to hypertonic stress primarily through the cytosolic accumulation of small uncharged molecules called organic osmolytes [5]. The concentration of organic osmolytes track extracellular osmolarity to maintain intracellular water content and cell volume. Additionally, through their chemical chaperone activity, osmolytes can also oppose the protein misfolding and aggregation that is a consequence of hypertonic stress [6]. Cells can accumulate hundreds of millimolar concentrations of organic osmolytes within hours of exposure to hypertonic stress. Osmolyte accumulation occurs either through the activity of specialized osmolyte transporters or osmolyte biosynthetic enzymes. In all cases, these transporters or biosynthesis enzymes are upregulated at the transcriptional and translational level by hypertonic stress [7]. The molecular identity of osmolyte transporters and biosynthetic enzymes utilized to accumulate osmolytes is highly variable between organisms and even between cells within the same organism. This is because there is a significant chemical diversity in osmolytes among bacteria, plants, and animals due to metabolic, nutritional, and ecological limitations [5].

One chemical class of osmolytes is carbohydrate polyols such as sorbitol and glycerol. During hypertonic stress, mammalian kidney epithelial cells upregulate the enzyme aldose reductase to synthesize sorbitol from glucose [8]. Likewise, *C. elegans* upregulates the biosynthetic enzyme glycerol-3-phosphate dehydrogenase (*gpdh-1*) to synthesize glycerol from glucose during exposure to hypertonic stress [9, 10]. Both sorbitol and glycerol accumulation provide osmoprotective effects, such as increased cellular volume, decreased intracellular ionic strength, and improved protein homeostasis. At the organismal level in *C. elegans*, decreased glycerol biosynthesis is associated with decreased fecundity and growth under hypertonic conditions [10]. In addition to osmolyte accumulation genes, hundreds of other genes are also upregulated during hypertonic stress [11]. While some of the transcriptional mechanisms leading to upregulation of these genes are known, post-transcriptional regulatory mechanisms are poorly understood [11, 12].

The *O*-GlcNAc transferase OGT is the sole protein that adds the single ring sugar, *O*-GlcNAc, to serine and threonine residues of hundreds of intracellular proteins to modify their function, stability, and localization. The *O*-GlcNAcase OGA is the sole enzyme that removes *O*-GlcNAc from proteins. OGT and OGA together regulate cellular *O*-GlcNAc homeostasis,

which is important to a variety of cellular processes including metabolism, stress responses, and proteostasis [13, 14]. Importantly, *O*-GlcNAc catalytic activity is not the only function of OGT. OGT proteolytically cleaves and activates the mammalian cell cycle regulator host cell factor C1 (HCF-1) [15, 16]. However, OGT-dependent HCF-1 cleavage does not occur in *C. elegans* or *Drosophila* [17, 18]. OGT also has non-catalytic functions in epithelial adherence junctions and an EEl-1-dependent E3 ubiquitin ligase complex in *C. elegans* GABA neurons [19, 20].

All metazoans express a single *ogt* gene, which is absent from yeast [21, 22]. Knockout of OGT in most metazoans is lethal at either the single cell or developmental level. The notable exception to this is *C. elegans*, where *ogt-1* null mutants are viable under standard cultivation conditions. Here, we show that *C. elegans ogt-1* mutants are non-viable under a specific physiological condition, hypertonic stress. Through an unbiased screen, we identified *ogt-1* as being required for expression of the osmosensitive *gpdh-1p::GFP* reporter. We found that under hypertonic stress conditions, *ogt-1* is required for accumulation of GPDH-1-GFP protein, but not *gpdh-1* mRNA. Additionally, *ogt-1* mutants are unable to develop following exposure to mild hypertonic environments. Finally, we demonstrate that expression of human OGT can partially rescue the *C. elegans* hypertonic stress phenotype. The ability of either human or *C. elegans* OGT to rescue is independent of O-GlcNAcylation catalytic activity but dependent on the OGT TPR domain. These results demonstrate for the first time a specific role for OGT in the essential process of osmoregulation and suggest that this function is conserved across >700 million years of evolution.

## Results

### An unbiased forward genetic screen for 'no induction of osmolyte biosynthesis gene expression' (Nio) mutants identifies the conserved *O*-GlcNAc transferase *ogt-1*

In *C. elegans*, hypertonic stress rapidly and specifically upregulates expression of the osmolyte biosynthesis gene *gpdh-1*, which we visualized with a *gpdh-1p::GFP* transcriptional reporter [10, 11]. To optimize this reporter for genetic screening, we added a *col-12p:dsRed* reporter, whose expression is not affected by hypertonic stress and serves as an internal control for non-specific effects on gene expression [23]. This dual reporter strain (*drIs4*) expresses only dsRed under isotonic conditions and both dsRed and GFP under hypertonic conditions, with very few animals exhibiting an intermediate phenotype (Fig 1A, 1B and 1C). A *gpdh-1p::GPDH-1-GFP* translational reporter (*kbIs6*) is also upregulated by hypertonic stress, but exhibits more variability than the *drIs4* transcriptional reporter (Fig 1D, 1E and 1F).

Taking advantage of the binary nature of GFP activation by hypertonic stress in the *drIs4* strain, we designed an unbiased F2 forward genetic screen for mutants that fail to activate GFP expression during hypertonic stress, but exhibit no effects on RFP (<u>N</u>o <u>i</u>nduction of <u>o</u>smolyte biosynthesis gene expression or Nio mutants; Fig 2A). From this screen of ~120,000 haploid genomes, we identified two recessive alleles, *dr15* and *dr20*, that genetically fail to complement each other. Whole genome sequencing and bioinformatics revealed that each allele contained a distinct nonsense mutation in the gene encoding the *O*-GlcNAc transferase *ogt-1* (S1, S2 and S3 Tables, Fig 2B). Two independently isolated *ogt-1* deletion alleles, *ok430* and *ok1474*, as well as wild type worms exposed to *ogt-1(RNAi)* also exhibited a Nio phenotype, and *ok430* and *ok1474* failed to complement the *dr15* and *dr20* alleles. (Fig 2B, 2C, 2D, S1A, S1B and S2 Tables). CRISPR reversion of the *dr20* Q600STOP mutation back to wild type was sufficient to rescue the *ogt-1* Nio phenotype, indicating that other ENU induced mutations in the background do not contribute to the Nio phenotype (Fig 2E). Additionally, transgenic

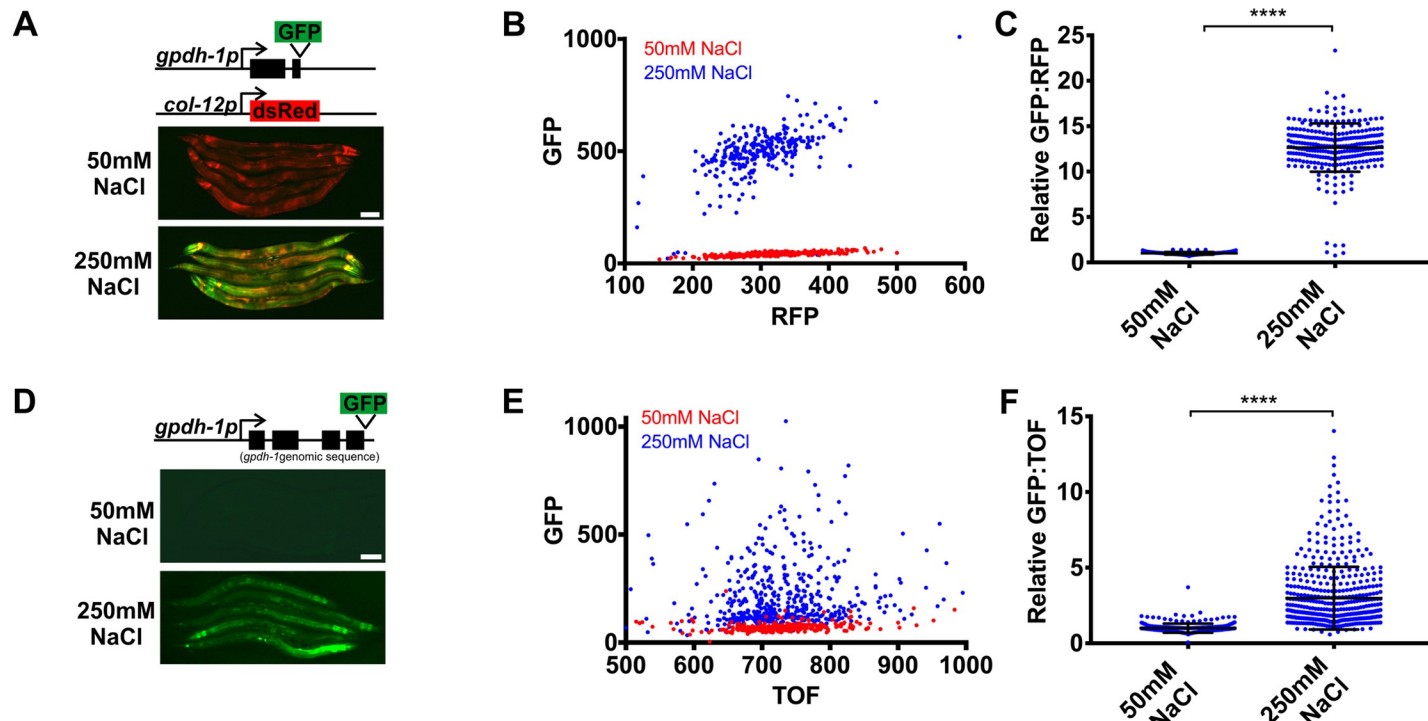

**Fig 1. *gpdh-1* transcriptional and translational reporters are upregulated by hypertonic stress.** (A) Wide-field fluorescence microscopy of day 2 adult animals expressing *drIs4* (*col-12p::dsRed*; *gpdh-1p::GFP*) exposed to 50 or 250 mM NaCl NGM plates for 18 hours. Images depict merged GFP and RFP channels. Scale bar = 100 microns. (B) COPAS Biosort quantification of GFP and RFP signal in day 2 adult animals expressing *drIs4* exposed to 50 or 250 mM NaCl NGM plates for 18 hours. Each point represents the quantified signal from a single animal. N ≥ 276 for each group. (C) Population mean of the normalized GFP/RFP ratio from data in 1B. Data are expressed as mean ± S.D. with individual points shown. ****—p<0.0001 (Mann-Whitney test). (D) Wide-field fluorescence microscopy of day 2 adult animals expressing a *kbIs6* (*gpdh-1p::GPDH-1-GFP)* translational fusion protein exposed to 50 or 250 mM NaCl NGM plates for 18 hours. Scale bar = 100 microns. (E) COPAS Biosort quantification of GFP and TOF signal in day 2 adult animals expressing the *kbIs6* translational fusion protein exposed to 50 or 250 mM NaCl NGM plates for 18 hours. N ≥ 276 for each group. (F) Population mean of the normalized GFP/TOF ratio from data in 1E. Data are expressed as mean ± S.D. ****—p<0.0001 (Mann-Whitney test).

overexpression of *ogt-1* in the *ogt-1(dr20)* mutant led to supra-physiological rescue (S1C Fig). Finally, we found that knock down of *ogt-1* during post-developmental stages with *ogt-1 (RNAi)* was sufficient to cause a Nio phenotype, suggesting that OGT-1 is not required for the establishment of developmental structures necessary for responding to hypertonic stress (Fig 2F). *ogt-1* is not required for the activation of other stress inducible reporters since inhibition of *ogt-1* resulted in small but significant increase in a heat shock inducible GFP reporter and had no effect on an endoplasmic reticulum stress inducible GFP reporter (S2 Fig). A reporter for the expression of the antimicrobial peptide *nlp-29* (*frIs7*) is induced by several stressors including hypertonic stress [23]. *ogt-1(RNAi)* did not prevent *nlp-29p::GFP* upregulation by hypertonicity and in some cases led to small but significant increase in the hypertonic induction of this reporter (S2E Fig) [23]. In conclusion, these results suggest that *ogt-1* is acutely required for hypertonic stress-induced upregulation of *gpdh-1p::GFP* reporter expression.

## OGT-1 is required for osmosensitive GPDH-1-GFP protein expression, but not osmosensitive transcription

Since *ogt-1* is required for induction of the *gpdh-1p::GFP* transgenic reporter by hypertonic stress, we hypothesized that endogenous osmosensitive mRNAs would not be upregulated in an *ogt-1* mutant. To test this, we used qPCR to measure the expression levels of several

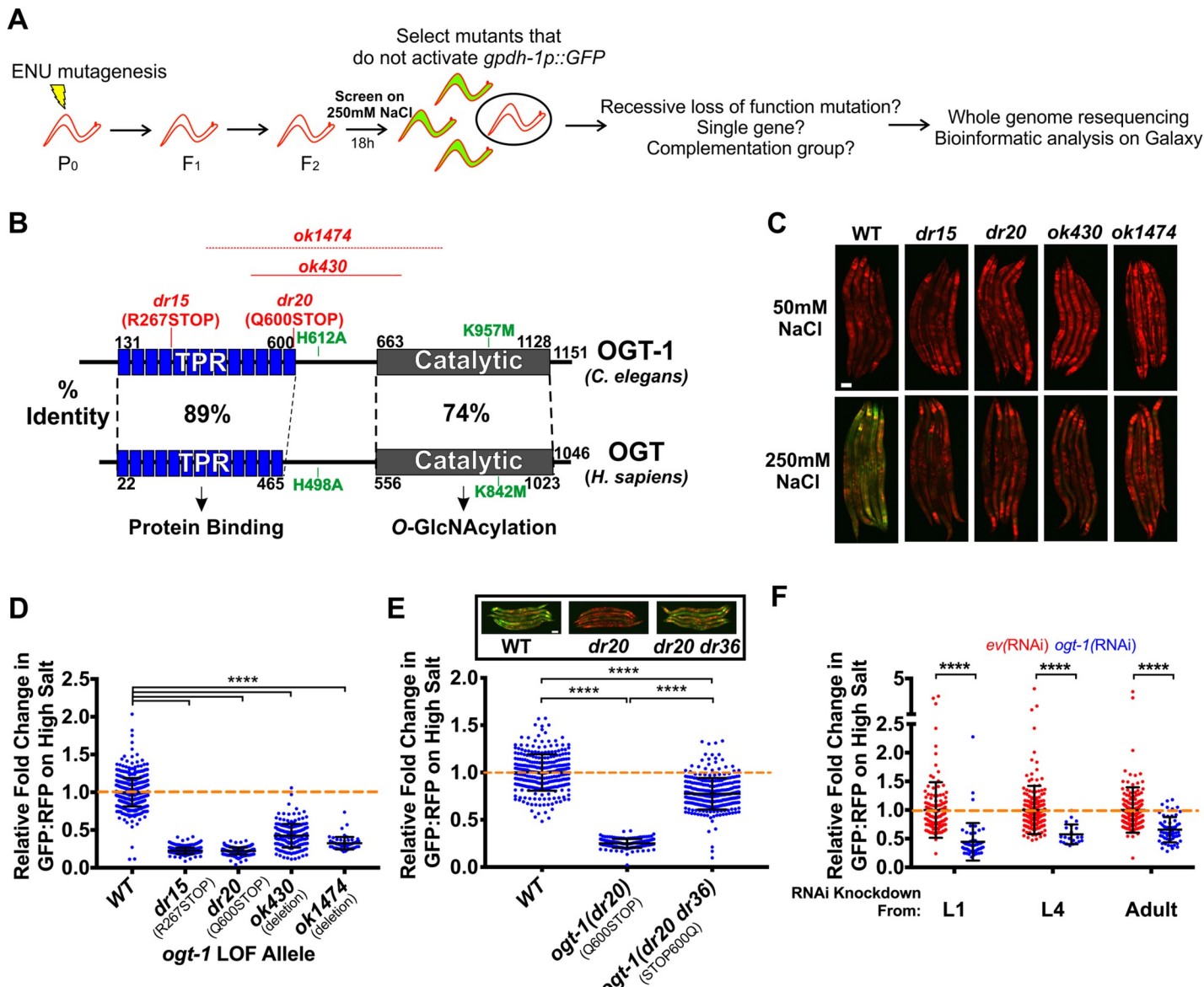

**Fig 2. The conserved *O*-GlcNAc transferase OGT-1 is required for the upregulation of the *gpdh-1* transcriptional reporter by hypertonic stress.** (A) ENU-based forward genetic screening strategy and mutant identification workflow. (B) *C. elegans* OGT and *Homo sapiens* OGT protein domain diagrams detailing the positions of the two LOF *ogt-1* alleles identified in the screen (*dr15* and *dr20*), two independently isolated *ogt-1* deletion mutations (*ok430* and *ok1474*), and two mutations that disrupt catalytic activity of the enzyme (H612A and K957M). The precise breakpoints of *ok1474* have not been determined. (C) Wide-field fluorescence microscopy of day 2 adult *drIs4* and *ogt-1;drIs4* mutant animals exposed to 50 or 250 mM NaCl NGM plates for 18 hours. Images depict merged GFP and RFP channels. Scale bar = 100 microns. (D) COPAS Biosort quantification of GFP and RFP signal in day 2 adult animals expressing *drIs4* or *ogt-1;drIs4* exposed to 50 or 250 mM NaCl NGM plates for 18 hours. Data are represented as the relative fold induction of normalized GFP/RFP ratio on 250 mM NaCl NGM plates versus 50 mM NaCl NGM plates, with WT fold induction set to 1. Each point represents the quantified signal from a single animal. Data are expressed as mean ± S.D. ****—p<0.0001 (Kruskal-Wallis test with post hoc Dunn's test). N ≥ 62 for each group. (E) COPAS Biosort quantification of GFP and RFP signal in day 2 adult animals expressing *drIs4* or *drIs4; ogt-1(dr20)* exposed to 50 or 250 mM NaCl NGM plates for 18 hours. *ogt-1(dr20 dr36)* is a strain in which the *dr20* mutation is converted back to WT using CRISPR/ Cas9 genome editing. Data are represented as relative fold induction of normalized GFP/RFP ratio on 250 mM NaCl NGM plates versus 50 mM NaCl NGM plates, with WT fold induction set to 1. Each point represents the quantified signal from a single animal. Data are expressed as mean ± S.D. ****—p<0.0001 (Kruskal-Wallis test with post hoc Dunn's test). N ≥ 170 for each group. *Inset*: Wide-field fluorescence microscopy of day 2 adult animals expressing *drIs4* in the WT or indicated *ogt-1* mutant background exposed to 250 mM NaCl NGM plates for 18 hours. Images depict merged GFP and RFP channels. Scale bar = 100 microns. (F) COPAS Biosort quantification of GFP and RFP signal in day 2 adult animals expressing *drIs4* exposed to 50 or 250 mM NaCl NGM plates for 18 hours. Animals were placed on *empty vector(RNAi) (ev(RNAi))* or *ogt-1(RNAi)* plates at the indicated stage. Data are represented as normalized fold induction of normalized GFP/RFP ratio on 250 mM NaCl RNAi plates relative to on 50 mM NaCl RNAi plates, with *ev(RNAi)* set to 1 for each RNAi timepoint. Each point represents the quantified signal from a single animal. Data are expressed as mean ± S.D. ****—p<0.0001 (Mann-Whitney test). N ≥ 144 for each group.

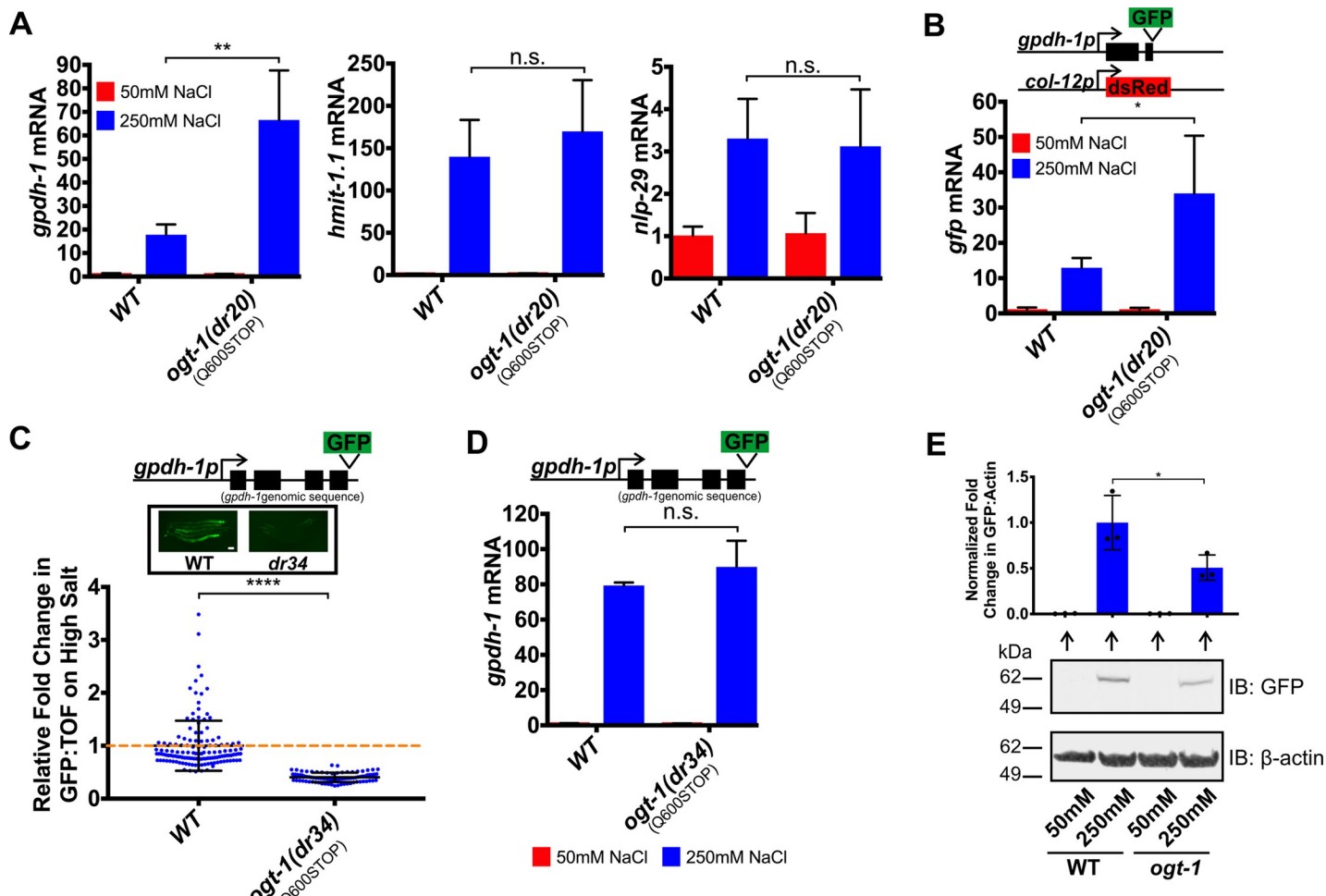

**Fig 3. OGT-1 functions post-transcriptionally to regulate osmosensitive GPDH-1-GFP protein expression.** (A) qPCR of *gpdh-1*, *hmit-1.1*, and *nlp-29* mRNA from WT and *ogt-1(dr20)* day 2 adult animals expressing *drIs4* exposed to 50 or 250 mM NaCl NGM plates for 24 hours. Data are represented as fold induction of RNA on 250 mM NaCl relative to 50 mM NaCl. Data are expressed as mean ± S.D. **—p<0.01, n.s. = nonsignificant (Student's two-tailed t-test). N ≥ 3 biological replicates of 35 animals for each group. (B) qPCR of *GFP* mRNA from WT and *ogt-1(dr20)* day 2 animals expressing *drIs4* exposed to 50 or 250 mM NaCl NGM plates for 24 hours. Data are represented as fold induction of RNA on 250 mM NaCl relative to 50 mM NaCl. Data are expressed as mean ± S.D. *—p<0.05 (Student's two-tailed t-test). N ≥ 3 biological replicates of 35 animals for each group. (C) COPAS Biosort quantification of GFP and TOF signal in day 2 adult animals expressing the *kbIs6* GPDH-1 translational fusion exposed to 50 or 250 mM NaCl NGM plates for 18 hours. The *ogt-1(dr34)* allele carries the same homozygous Q600STOP mutation as the *ogt-1 (dr20)* allele and was introduced using CRISPR/Cas9. Data are represented as relative fold induction of normalized GFP/TOF ratio on 250 mM NaCl NGM plates versus 50 mM NaCl NGM plates. Each point represents the quantified signal from a single animal. Data are expressed as mean ± S.D. ****—p<0.0001 (Mann-Whitney test). N ≥ 84 for each group. *Inset*: Wide-field fluorescence microscopy of day 2 adult animals expressing the *kbIs6* translational fusion protein exposed to 250 mM NaCl NGM plates for 18 hours. Scale bar = 100 microns. (D) qPCR of *gpdh-1* mRNA from day 2 adult animals expressing the *kbIs6* translational fusion exposed to 50 or 250 mM NaCl NGM plates for 24 hours. Strains include WT and *ogt-1(dr34)*. The *ogt-1(dr34)* allele is the *dr20* point mutation introduced using CRISPR/Cas9. Data are represented as fold induction of RNA on 250 mM NaCl relative to 50 mM NaCl. Data are expressed as mean ± S.D. n.s. = nonsignificant (Student's two-tailed t-test). N = 3 biological replicates of 35 animals for each group. (E) Immunoblot of GFP and β-actin in lysates from day 2 adult animals exposed to 50 mM or 250 mM NaCl for 18 hours. The animals express a CRISPR/Cas9 edited knock-in of GFP into the endogenous *gpdh-1* gene (*gpdh-1(dr81)*). *ogt-1* carries the *dr83* allele, which is the same homozygous Q600STOP mutation as the *ogt-1(dr20)* allele and was introduced using CRISPR/Cas9. *Top*: Normalized quantification of immunoblots. *—p<0.05 (One-way ANOVA with post hoc Dunnett's test). *Bottom*: Representative immunoblot. N = 3 biological replicates.

previously described mRNAs that are induced by osmotic stress [11]. Surprisingly, we found that osmotically induced mRNA expression of *gpdh-1*, *nlp-29*, and *hmit-1.1* was still upregulated in *ogt-1(dr20)* (Fig 3A and S3A Fig). Consistent with this, we also observed that GFP mRNA derived from the overexpressed *gpdh-1p::GFP* reporter *drIs4* was upregulated by hypertonic stress in *ogt-1(dr20)* even though GFP protein levels were strongly reduced (Figs 3B and

2D, S1A Fig). These data unexpectedly suggest that OGT-1 regulates hypertonicity-induced GPDH-1-GFP protein expression at a post-transcriptional level.

To further examine if OGT-1 affects the coupling between hypertonic stress induced mRNA and protein expression, we measured GPDH-1-GFP protein levels in an *ogt-1* mutant (*dr34*; CRISPR/Cas9 knock-in of the *dr20*(Q600STOP) mutation) expressing a GPDH-1 translational reporter (GPDH-1::GFP). As we observed for the *gpdh-1p*::*GFP* transcriptional reporter, *ogt-1(dr34)* mutants failed to induce the GPDH-1::GFP protein in response to hypertonic stress (Fig 3C, S3B Fig). However, the mRNA from this translational reporter was still induced to wild type levels (Fig 3D). mRNA induction of the translational reporter did not exceed wild type levels like we saw for the transcriptional reporter for unknown reasons. Importantly the requirement for *ogt-1* in the hypertonic stress response is not transgene dependent because *ogt-1* is also required for the hypertonic induction of a CRISPR/Cas9 engineered endogenously expressed GPDH-1::GFP fusion protein, which we confirmed to be functional based on its ability to exhibit acute adaptation to hypertonic stress (Fig 3E and S4D Fig). Like in the transcriptional and translational *gpdh-1* reporters, *gpdh-1*::*gfp* mRNA levels in the *gpdh-1*::*gfp* CRISPR allele were induced to WT levels or higher (S3C and S3D Fig). In conclusion, these results suggest that *ogt-1* functions downstream of osmosensitive mRNA upregulation, but upstream of osmosensitive GPDH-1-GFP protein expression.

## Physiological and genetic adaptation to hypertonic stress requires *ogt-1*

*C. elegans* upregulates osmosensitive genes, including *gpdh-1*, to survive and adapt to hypertonic challenges. Survival and adaptation can be measured in several ways. Survival measures the ability of animals grown under standard laboratory isotonic conditions to survive a 24 hour exposure to an indicated level of hypertonic stress. Acute adaptation measures the ability of animals to activate adaptive responses that permit survival under normally lethal hypertonic conditions using a pre-conditioning stimulus. Chronic adaptation measures the ability of animals develop under non-lethal hypertonic conditions.

We found that loss of *ogt-1* had no effect on acute survival during hypertonic stress (S4A Fig) [10]. However, loss of *ogt-1* blocked both acute and chronic adaptation to hypertonic stress (Fig 4A and S4B Fig). Chronic adaptation did not alter *ogt-1(dr20)* egg laying under hypertonic conditions (mean +/- S.D. of total eggs laid on 250 mM NaCl–wild type 35.8 +/- 27.4; *ogt-1(dr20)* 27.9 +/- 21.6, n = 10 for each genotype, p = 0.92, 2-tailed Student's T-test). However, embryo development under hypertonic conditions, but not isotonic conditions, was inhibited in *ogt-1(dr20)* (S4C Fig). The acute adaptation phenotype was rescued by CRISPR reversion of the *dr20* Q600STOP mutation to wild type (S4D Fig). Interestingly, the acute adaptation phenotype of *ogt-1* mutants must extend beyond its effects on *gpdh-1*, since the acute adaptation phenotype of a *gpdh-1* presumptive null mutant is not as severe as that observed in an *ogt-1* mutant (Fig 4A).

In addition to physiological exposures, adaptation to hypertonic stress can also be induced genetically via loss of function mutations in several hypodermis expressed secreted extracellular matrix (ECM) proteins [11, 12]. These mutants exhibit maximal induction of *gpdh-1* mRNA and accumulation of glycerol. As a result, these mutants are constitutively adapted to survive normally lethal levels of hypertonic stress [11, 12]. To test if *ogt-1* is required for genetic adaptation to hypertonic stress, we introduced an *ogt-1* mutation into *osm-8(dr9)* or *osm-11(n1604)* mutants. Both *osm-8* and *osm-11* mutants exhibit constitutively elevated *gpdh-1p*::*GFP* expression under isotonic conditions. However, *gpdh-1p*::*GFP* levels were significantly reduced in *osm-8;ogt-1* and *osm-11;ogt-1* double mutants (Fig 4B and S4E Fig). Consistent with this observation, the ability of *osm-8* mutants to survive a lethal hypertonic stress was

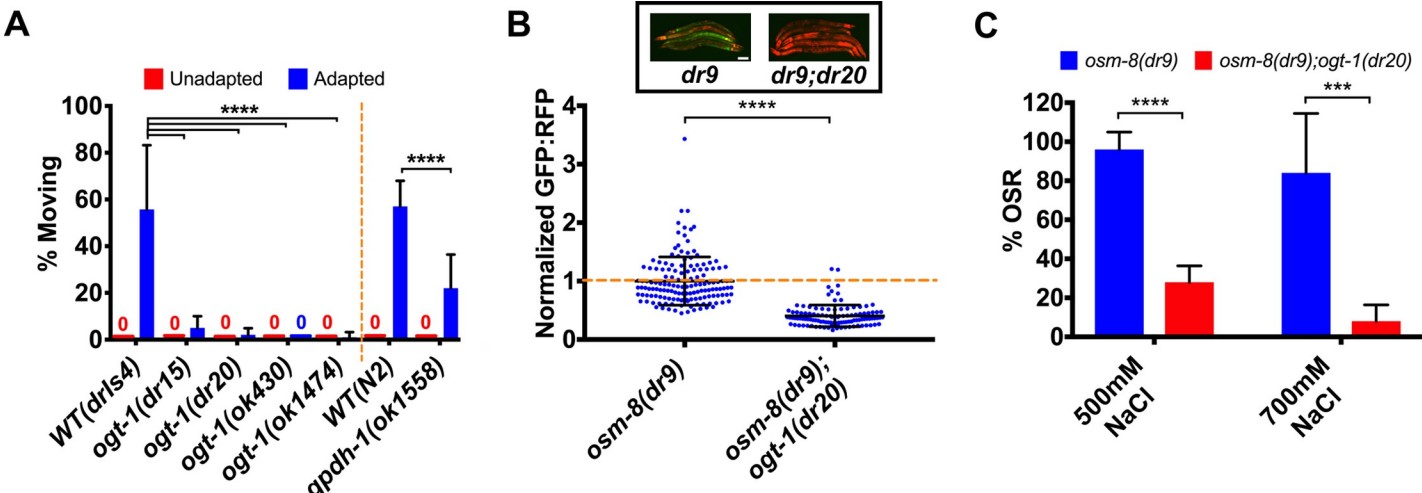

**Fig 4. *ogt-1* is required for physiological and genetic adaptation to hypertonic stress.** (A) Percent of moving unadapted and adapted day 3 adult animals exposed to 600 mM NaCl NGM plates for 24 hours. Strains expressing *drIs4* are on the left of the dashed orange line and those not expressing *drIs4* are on the right. *ok1558* is an out-of-frame deletion allele that generates a premature stop codon in exon 2 of *gpdh-1* and is therefore a likely null allele. Data are expressed as mean ± S.D. ****— $p < 0.0001$ (One-way ANOVA with post hoc Dunnett's test). N = 5 replicates of 20 animals for each strain. (B) COPAS Biosort quantification of GFP and RFP signal in day 2 adult animals expressing *drIs4* exposed to 50 mM NaCl NGM plates. Data are represented as the fold induction of normalized GFP/RFP ratio on 50 mM NaCl NGM plates, with *osm-8(dr9)* set to 1. *osm-8(dr9)* was isolated in a previous genetic screen for new *osm-8* alleles but encodes the same mutation as the *n1518* reference allele. Each point represents the quantified signal from a single animal. Data are expressed as mean ± S.D. ****— $p < 0.0001$ (Mann-Whitney test). N ≥ 109 for each group. *Inset*: Wide-field fluorescence microscopy of day 2 adult animals expressing *drIs4* exposed to 50 mM NaCl NGM plates. Images depict merged GFP and RFP channels. Scale bar = 100 microns. (C) Percent of moving (OSR, osmotic stress resistant) day 1 animals after exposure to 500 mM NaCl or 700 mM NaCl for 10 minutes. Data are represented as mean ± S.D. ***— $p < 0.001$, ****— $p < 0.0001$ (Student's two-tailed t-test). N = 5 replicates of 10 animals for each strain.

suppressed in the *osm-8;ogt-1* double mutants (Fig 4C) [12]. These data suggest that *ogt*-1 is required for both physiological and genetic adaptation to hypertonic stress caused by loss of the ECM proteins OSM-8 and OSM-11.

## Non-canonical activity of *ogt-1* in the hypodermis regulates *gpdh-1* induction by hypertonic stress through a functionally conserved mechanism

In *C. elegans*, a CRISPR generated OGT-1-GFP allele is functional and is ubiquitously expressed throughout somatic cells in the nucleus, consistent with previous observations (S4D and S5D Figs) [24]. Therefore, we used tissue specific promoters to test which tissues require *ogt-1* expression for *gpdh-1* induction by hypertonic stress. The expression of *ogt-1* from either its native promoter or a hypodermal specific promoter was sufficient to rescue *gpdh-1* induction by hypertonic stress in *ogt-1* LOF mutants (Fig 5A and 5B and S5A Fig). Expression of *ogt-1* from an intestinal specific promoter or a muscle specific promoter caused weak but statistically significant rescue (Fig 5A and 5B and S5A Fig). Expression of *ogt-1* in the neurons did not rescue (Fig 5A and 5B and S5A Fig). Since *gpdh-1* is induced by hypertonic stress in the hypodermis [10], these results suggest that *ogt-1* primarily acts cell autonomously in the hypoderm to regulate osmosensitive protein expression.

Given that *C. elegans* OGT-1 is highly conserved with human OGT (Fig 2B), we asked if human OGT could functionally replace *C. elegans* OGT-1 in the hypertonic stress response. Overexpression of a human OGT cDNA from the native *C. elegans ogt-1* promoter exhibited weak but statistically significant rescue of *gpdh-1p::GFP* induction by hypertonic stress in an *ogt-1* LOF mutant (Fig 5C and 5D and S5B Fig). Unexpectedly, catalytically inhibited human OGT (OGT H498A) rescued *gpdh-1p::GFP* induction by hypertonic stress in an *ogt-1(dr20)*

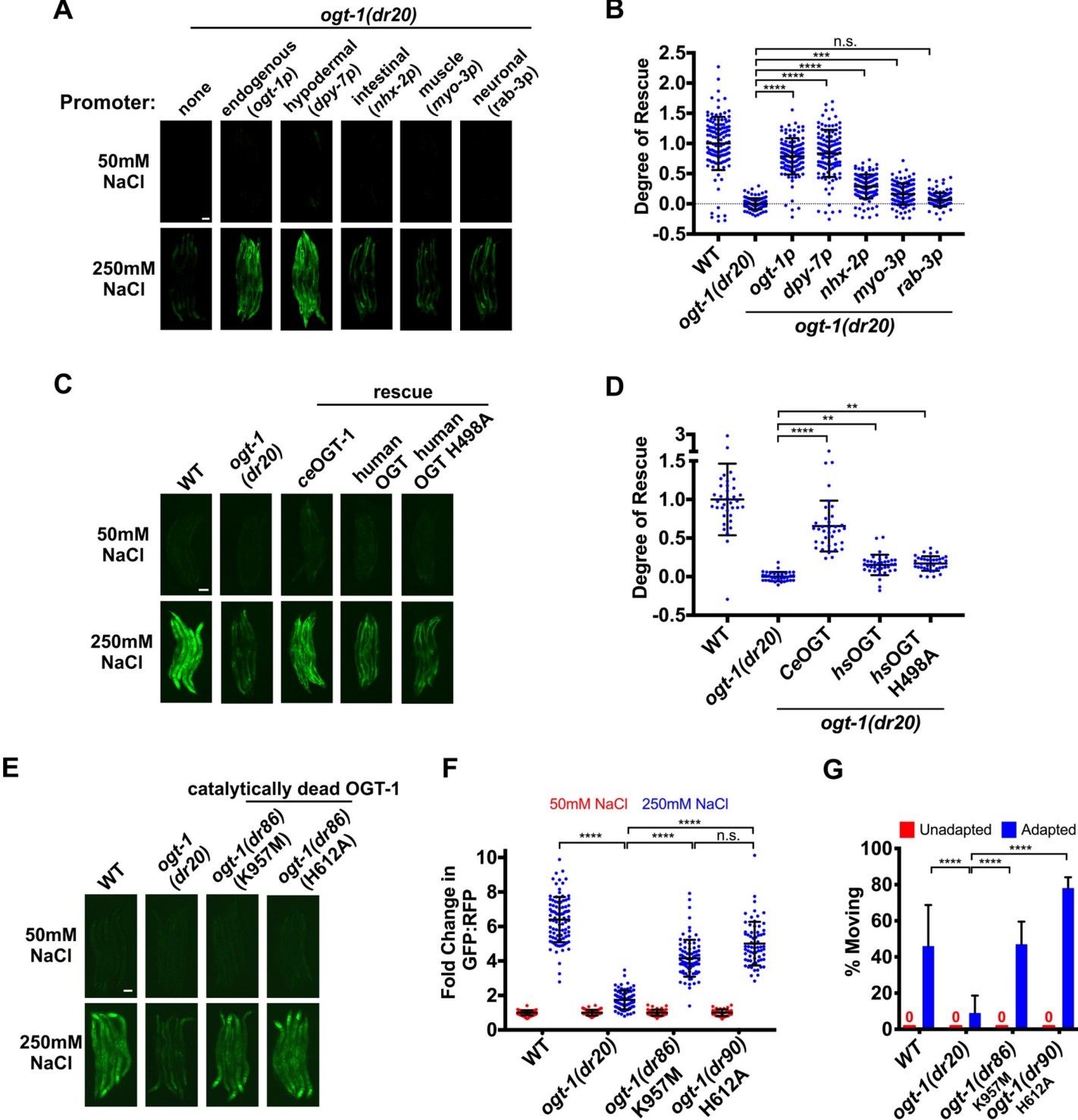

**Fig 5. Non-canonical activity of *ogt-1* primarily in the hypodermis regulates *gpdh-1* induction by hypertonic stress through a functionally conserved mechanism.**
(A) Wide-field fluorescence microscopy of day 2 adult animals expressing *drIs4* exposed to 50 or 250 mM NaCl NGM plates for 18 hours. Strains express an *ogt-1* cDNA from the indicated tissue-specific promoter. Images depict the GFP channel only for clarity. The RFP signal was unaffected in these rescue strains (not shown). Scale bar = 100 microns. (B) COPAS Biosort quantification of GFP and RFP signal in day 2 adult animals expressing *drIs4* exposed to 50 or 250 mM NaCl NGM plates for 18 hours. Data are represented as the 'Degree of Rescue' on 250 mM NaCl NGM plates relative to on 50 mM NaCl NGM plates (see 'Methods' for description of this calculation). Each point represents the quantified signal from a single animal. Data are expressed as mean ± S.D. ***—p<0.001, ****—p<0.0001, n.s. = nonsignificant (Kruskal-Wallis test with post hoc Dunn's test). N ≥ 110 for each group. (C) Wide-field fluorescence microscopy of day 2 adult animals expressing *drIs4* exposed to 50

or 250 mM NaCl NGM plates for 18 hours. For the WT and catalytically inactive human rescue strains, we expressed a human cDNA corresponding to isoform 1 of OGT using an extrachromosomal array. Images depict the GFP channel for clarity. The RFP signal was unaffected in these rescue strains (not shown). Scale bar = 100 microns. (D) COPAS Biosort quantification of GFP and RFP signal in day 2 adult animals expressing *drIs4* exposed to 50 or 250 mM NaCl NGM plates for 18 hours. Data are represented as the 'Degree of Rescue' on 250 mM NaCl NGM plates relative to on 50 mM NaCl NGM plates (see 'Methods' for description of this calculation). Each point represents the quantified signal from a single animal. Data are expressed as mean ± S.D. ****—p<0.0001, ***—p <0.001, **—p<0.01 (Kruskal-Wallis test with post hoc Dunn's test). N ≥ 40 for each group. (E) Wide-field fluorescence microscopy of day 2 adult animals expressing *drIs4* exposed to 50 or 250 mM NaCl NGM plates for 18 hours. Images depict the GFP channel only for clarity. The RFP signal was unaffected in these rescue strains (not shown). Scale bar = 100 microns. (F) COPAS Biosort quantification of GFP and RFP signal in day 2 adult animals expressing *drIs4* exposed to 50 or 250 mM NaCl NGM plates for 18 hours. Data are represented as the fold induction of normalized GFP/RFP ratio on 50 and 250 mM NaCl NGM plates relative to on 50 mM NaCl NGM plates. Each point represents the quantified signal from a single animal. Data are expressed as mean ± S.D. ****—p<0.0001, n.s = nonsignificant (Kruskal-Wallis test with post hoc Dunn's test). N ≥ 81 for each group. (G) Percent of moving unadapted and adapted day 3 adult animals expressing *drIs4* exposed to 600 mM NaCl NGM plates for 24 hours. Data are expressed as mean ± S.D. ****—p<0.0001 (One-way ANOVA with post hoc Tukey's test). N = 5 replicates of 20 animals for each strain.

LOF mutant to the same extent as wild type human OGT (Fig 5C and 5D and S5B Fig) [25, 26]. To further test the requirement for OGT-1 *O*-GlcNAcylation in the hypertonic stress response, we CRISPR engineered catalytically inactive mutations into the endogenous *C. elegans ogt-1* locus (H612A and K957M, equivalent to human H498A and K842M) [20, 26]. Surprisingly, only the K957M mutation suppressed *O*-GlcNAcylation activity completely (S5C Fig). The H612A mutation reduced *O*-GlcNAcylation but did not eliminate it (S5C Fig). However, neither the K957M nor the H612A mutation altered OGT-1 protein levels or nuclear localization (S5D Fig). In agreement with the results from the catalytically inhibited human OGT, *C. elegans* expressing catalytically impaired alleles of endogenous *ogt-1* induced *gpdh-1p*::*GFP* during hypertonic stress and had normal adaptation to hypertonic stress (Fig 5E, 5F and 5G). In conclusion, a non-catalytic function of OGT-1 in the hypodermis is required for osmosensitive protein induction by hypertonic stress and this function may be conserved from *C. elegans* to humans.

In addition to the catalytic domain of OGT, another functionally important domain in OGT is the tetratricopeptide repeat (TPR) domain. This domain mediates protein-protein interactions thought to be important for the binding of *O*-GlcNAcylation substrates [25, 27, 28]. To determine if the TPR domain is also required for the *O*-GlcNAcylation-independent hypertonic stress response in *C. elegans*, we used CRISPR to engineer a complete deletion of the TPR domain in the *ogt-1* locus (ΔTPR(128 aa– 583 aa)). The ΔTPR mutation suppressed *O*-GlcNAcylation completely (S6A Fig). However, it did not alter OGT-1::GFP protein levels or localization (S6B Fig). *C. elegans* expressing the *ogt-1* ΔTPR allele (*ogt-1(dr93)*) had impaired *gpdh-1p*::*GFP* induction during hypertonic stress and impaired adaptation during hypertonic stress (S6C, S6D and S6E Fig). Therefore, a TPR-dependent, *O*-GlcNAcylation-independent function of OGT-1 is required for the hypertonic stress response.

## Discussion

Through an unbiased forward genetic screen for mutants that disrupt osmosensitive expression of a *gpdh-1*::*GFP* reporter in *C. elegans*, we identified multiple alleles of the *O*-GlcNAc transferase OGT-1. *ogt-1* likely functions as a key signaling component of the hypertonic stress response, since post-developmental knockdown of *ogt-1* is sufficient to cause the Nio phenotype. *ogt-1*-dependent signaling in the hypertonic stress response appears to occur primarily in the hypodermis, a known osmosensitive tissue in *C. elegans* [10, 12]. The mechanism by which *ogt-1* regulates hypertonicity induced gene expression is unexpected. *ogt-1* mutants exhibit normal hypertonicity induced upregulation of stress response mRNAs. However, the level of at least one reporter protein, GPDH-1::GFP, is significantly reduced, suggesting *ogt-1* acts via a post-transcriptional mechanism(s). Interestingly, *ogt-1* is not required for the hypertonic induction of the *nlp-29p*::*GFP reporter*, suggesting that the effect of *ogt-1* is linked to *gpdh-1*

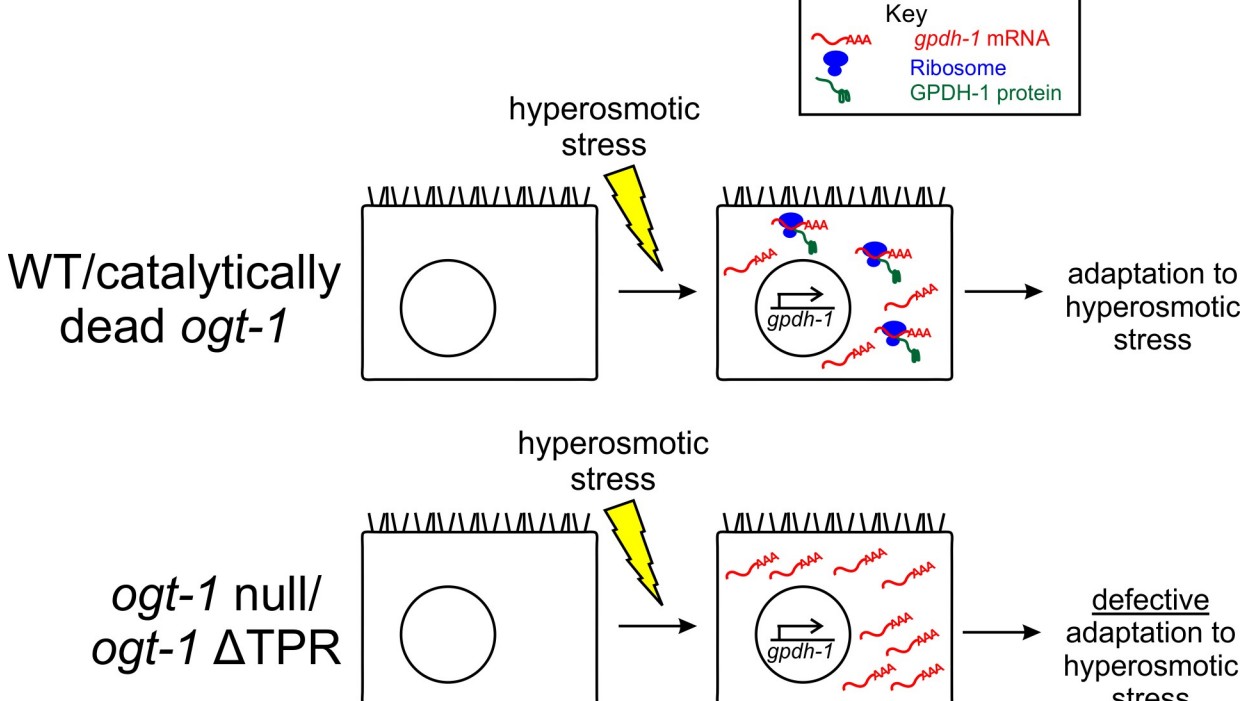

**Fig 6. A non-catalytic function of *ogt-1* is required to couple hypertonic stress induced transcription and translation to enable physiological adaptation to hypertonic stress.** In WT, animals exposed to hypertonic stress induce the transcription of osmosensitive mRNAs, such as *gpdh-1*. These mRNAs are rapidly translated into protein by the ribosome, facilitating adaptation to hyperosmotic stress. Loss of *ogt-1* does not interfere with hypertonic stress induced transcription. Rather, loss of *ogt-1* decreases hypertonic stress induced protein levels. *ogt-1* may facilitate stress-induced translation via several potential mechanisms, including regulation of mRNA cleavage and 3'UTR usage, mRNA export, initiation factor interactions, or ribosomal elongation of the transcript. Importantly, the tetratricopeptide repeat (TPR) domain but not the *O*-GlcNAcylation function of OGT-1 is required in the hypertonic stress response.

and not GFP. *ogt-1* mutants are completely unable to adapt and develop in hypertonic environments and this correlates with an inability of *ogt-1* mutants to properly upregulate the translation of osmoprotective proteins such as GPDH-1-GFP. Intriguingly, we demonstrate that the function of *ogt-1* in the hypertonic stress response does not require *O*-GlcNAcylation catalytic activity. Both wild type and catalytically inactive human OGT can rescue the *C. elegans ogt-1* Nio phenotype, suggesting that this non-catalytic function is also conserved with humans. However, *C. elegans* expressing OGT-1 without the tetratricopeptide repeat (TPR) domain are Nio, suggesting that the TPR domain is required for the hypertonic stress response (Fig 6).

*C. elegans* is the primary genetic model system for studies of *ogt-1* because it is the only organism in which loss of *ogt-1* is viable [29, 30]. This has allowed many previous studies to parse the roles of *ogt-1* in lifespan, metabolism, innate immunity, behavior, neuron function, stress responses, cell fate, and autophagy [20, 24, 29, 31–41]. Importantly, most of these studies utilized global *ogt-1* knockdown, which eliminates both O-GlcNAcylation-dependent and–independent functions of *ogt-1*. The missense alleles generated here will provide powerful tools for differentiating between these functions. Further structure-function studies are needed to determine if the hypertonic functions of *ogt-1* can be molecularly separated from its *O*-GlcNAcylation-dependent functions. If conserved, such mutations could provide important new insights into the physiological role of non-catalytic OGT functions in humans.

Our studies reveal a critical and previously unappreciated condition-specific role of OGT-1 in adaptation to hypertonic stress. This phenotype is completely penetrant and one of the

strongest *ogt-1* phenotypes described to date. Although their ability to survive acute hypertonic stress is unaffected, *ogt-1* mutants are unable to adapt and develop following extremely mild shifts in extracellular osmolarity (250 mM NaCl). Such conditions have minimal effects on the ability of wild type animals to adapt and develop [10]. This suggests a critical physiological role of *ogt-1* in the ability of *C. elegans* to survive in the wild, since worms are continuously exposed to fluctuating environmental salinity in their native ecosystems [42]. Given that both *C. elegans* and human OGT are able to rescue the Nio phenotype of *ogt-1* mutants in *C. elegans*, we speculate that OGT plays an ancient and conserved physiological function in response to environmental and physiological perturbations in osmotic homeostasis.

In mammals, OGT is essential for cell division, a physiological process that involves tight regulation of cell volume [30, 43–45]. Therefore, we speculate that OGT may be required for mammalian cell division for the same reason it is essential for adaptation to hypertonic stress in *C. elegans*: it plays a critical role in cell volume regulation. One reason mammalian cells may be unable to divide without OGT is because they cannot properly regulate cell volume during cell division. In *C. elegans*, unlike in mammals, *ogt-1* is not an essential gene. We hypothesize that the osmotic homogeneity of standard *C. elegans* lab culture conditions allows *ogt-1* mutants to survive and propagate normally. However, under hypertonic conditions, *ogt-1* becomes an essential gene in *C. elegans*, like it is in humans. It will be interesting to explore the roles and requirements of OGT in cell volume regulation in mammalian cells and tissues.

Knockout of OGT in mammalian cells leads to a rapid loss in cellular viability [30]. This phenotype is largely thought to be due to loss of *O*-GlcNAcylation activity. However, data from human cells suggest that *O*-GlcNAcylation activity may not be the essential function of OGT. For example, exposure of mammalian cells to the *O*-GlcNAc inhibitor $Ac_4$-5$S$GlcNAc largely blocks *O*-GlcNAcylation, but cellular viability and division are unaffected [46]. Additionally, cells and humans carrying inherited catalytic point mutations in OGT associated with intellectual disability are viable [47]. Our data show that the role of OGT-1 in the *C. elegans* hypertonic stress response is also independent of catalytic activity. Such catalytically-independent roles of OGT-1 have also been described in the context of synaptic regulation and cell adhesion [19, 20]. If the evolutionarily critical role of OGT in mammalian cells is related to its ability to regulate cell volume, our data suggest that such functions are independent of *O*-GlcNAcylation activity. These non-catalytic functions of OGT and the protein domains that regulate these functions are largely unexplored. The *C. elegans* Nio phenotype may provide a powerful genetic system for identifying new functional domains important for OGT function via targeted and unbiased genetic screening strategies.

Cell volume regulation during environmental stress requires upregulation of osmoprotective proteins, including those that regulate osmolyte accumulation. In almost all cases, these genes are upregulated at the transcriptional level [11]. Our findings are the first evidence that this pathway is also under post-transcriptional control. OGT-1 is required for the accumulation of a GFP tagged GPDH-1 protein during hypertonic stress, but not for the upregulation of *gpdh-1* mRNA. Interestingly, this is not a complete elimination of GPDH-1-GFP protein induction and even if it were, *gpdh-1* null mutants still retain significant hypertonic adaptation potential, whereas *ogt-1* mutants are completely adaptation deficient. Alternatively, OGT-1 may in fact regulate *gpdh-1* mRNA expression at a transcriptional level but only in a specific tissue, such as the hypodermis. Upregulation of *gpdh-1* in other tissues, like the intestine, may mask such tissue specific regulation. Nevertheless, our results suggest that OGT-1 regulation of the hypertonic stress response is likely to extend beyond its effects on GPDH-1 induction. The nature of these additional targets and/or mechanisms is currently unknown.

The regulation of stress responsive gene expression by OGT is not a new paradigm. Previous data has shown that it plays both a transcriptional and post-transcriptional role in stress

response gene expression. For example, OGT-1 *O*-GlcNAcylates the oxidative stress responsive transcription factor SKN-1 to facilitate upregulation of antioxidant gene transcription [36]. On the other hand, OGT regulates UPR^ER and HSR gene expression post-transcriptionally by *O*-GlcNAcylating translation initiation factors to selectively facilitate translation of stress induced mRNAs [48, 49]. Importantly, all of the previously described roles of OGT in stress responses require *O*-GlcNAcylation. While our data suggests that OGT-1 also functions in the hypertonic stress response through a post-transcriptional mechanism, this mechanism is fundamentally different from that of the oxidative, HSR, and UPR^ER stress responses because it does not require OGT *O*-GlcNAcylation activity [48, 49]. Further mechanistic studies are needed to define the *O*-GlcNAcylation-independent downstream targets and mechanisms of OGT-1 required for osmoprotective protein expression.

Since the discovery of OGT, *C. elegans* has been an important tool for characterizing the role of OGT in cell signaling because it is the only organism in which genetic loss of OGT generates viable cells and organisms [22, 29]. However, it is still unknown why *ogt-1* null *C. elegans*, in contrast to every other metazoan, is viable [50]. One possibility is that the evolutionarily conserved role of *ogt-1* in cell volume regulation during hypertonic stress contributes to the essential role of OGT in all metazoans, including *C. elegans*. However, several key questions about the osmoprotective nature of OGT still remain. First, while canonical OGT-1-dependent *O*-GlcNAcylation is dispensable for the hypertonic stress response, it remains unclear which functions of *ogt-1* are important to this physiological process. Although OGT-1 can also catalyze a unique type of proteolysis event, this activity is not thought to occur in *C. elegans* [18]. Regardless, the K957M mutation also eliminates the known proteolytic activity of *ogt-1*, suggesting that this function is also not required in the hypertonic stress response [26]. Future studies, utilizing both targeted *ogt-1* deletion alleles and unbiased genetic screens for new *ogt-1* missense mutations with a Nio phenotype, should help resolve this question. Second, the precise post-transcriptional mechanism under OGT-1-dependent control remains unknown. Such mechanisms could include mRNA cleavage and polyadenylation site usage, mRNA nuclear export, selective interactions between ribosomes and stress-induced mRNAs, or regulated proteolysis of stress-induced proteins such as GPDH-1. While most of these potential mechanisms await testing, we find that autophagic or proteasome-mediated proteolysis does not appear to be involved (S7 Fig). Finally, it remains unclear which genes *ogt-1* coordinates with to regulate hypertonic stress signaling. Future studies analyzing new Nio mutants should shed light on these interactions.

In conclusion, our unbiased genetic screening approaches in *C. elegans* have revealed a previously unappreciated requirement for non-canonical OGT signaling in a critical and conserved aspect of cell physiology. The primary function of OGT has long been assumed to be due to its catalytic *O*-GlcNAcylation activity. However, as we and others have shown, OGT also has critical and conserved non-catalytic functions that warrant further study [19, 20]. It is vital that future studies involving OGT utilize point mutants that differentiate canonical from non-canonical functions rather than OGT knockouts, which ablate both. As our studies have shown, such approaches could reveal new roles for this key protein in unexpected aspects of cell physiology.

## Materials and methods

### *C. elegans* strains and culture

Strains were cultured on standard NGM media with *E.coli* OP50 bacteria at 20˚C unless otherwise noted. The following strains were used; N2 Bristol WT, OG119 *drIs4* [*gpdh-1p::GFP; col-12p::dsRed2*], VP223 *kbIs6* [*gpdh-1p::gpdh-1-GFP*], OG971 *ogt-1(dr15);drIs4*, OG969 *ogt-1*

*(dr20);drIs4*, OG1034 *ogt-1(ok430);drIs4*, OG1035 *ogt-1(ok1474);drIs4*, OG1066 *ogt-1(dr20 dr36);drIs4*, OG1064 *ogt-1(dr34);unc-119(ed3);kbIs6*, OG1115 *gpdh-1(dr81)* [*gpdh1::GFP*], OG1123 *gpdh-1(dr81);ogt-1(dr84)*, RB1373 *gpdh-1(ok1558)*, OG1048 *osm-8(dr9);unc-4(e120); drIs4*, OG1049 *osm-8(dr9);unc-4(e120);ogt-1(dr20):drIs4*, OG1111 *ogt-1(dr20);drIs4;drEx468* [*ogt-1p::ogt-1cDNA::ogt-13'utr; rol-6(su1006)*], OG1119 *ogt-1(dr20);drIs4;drEx469* [*dpy-7p::ogt-1cDNA::ogt-13'utr; rol-6(su1006)*], OG1120 *ogt-1(dr20);drIs4;drEx470* [*nhx-2p::ogt-1cDNA:: ogt-13'utr; rol-6(su1006)*], OG1121 *ogt-1(dr20);drIs4;drEx471* [*myo-2p::ogt-1cDNA::ogt-13'utr; rol-6(su1006)*], OG1122 *ogt-1(dr20);drIs4;drEx472* [*rab-3p::ogt-1cDNA::ogt-13'utr; rol-6 (su1006)*], OG1125 *ogt-1(dr20);drIs4;drEx473* [*ogt-1p::human OGT isoform 1cDNA::ogt-13'utr; rol-6(su1006)*], OG1126 *ogt-1(dr20);drIs4;drEx474* [*ogt-1p::human OGT isoform 1 H498AcDNA::ogt-13'utr; rol-6(su1006)*], OG1046 *ogt-1(dr20);drIs4;drEx465* [*ogt-1p::ogt-1 genomic*], TJ375 *gpIs1* [*hsp16.2p::GFP*], SJ4005 *zcIs4* [*hsp4::GFP*] V, OG1081 *ogt-1(dr50);zcIs4*, MT3643 *osm-11(n1604)*, OG1083 *ogt-1(dr52);osm-11(n1604)*, OG1135 *ogt-1(dr86);drIs4*, OG1140 *ogt-1(dr90);drIs4*, OG1124 *ogt-1(dr84)* [*ogt-1*::GFP], OG1139 *ogt-1(dr84 dr89)*, OG1141 *ogt-1(dr84 dr91)*, OG1156 *ogt-1(dr93);drIs4*, OG1157 *ogt-1(dr84 dr94)*, IG274 *frIs7* [*nlp-29p::GFP + col-12p::dsRed]*. To create mutant combinations, we used either standard genetic crossing approaches or CRISPR/Cas9 genetic engineering (see below for CRISPR methods). The homozygous genotype of every strain was confirmed either by DNA sequencing of the mutant lesion, restriction digest, or a loss of function phenotype.

## Genetic methods

**ENU mutagenesis and mutant isolation.**   L4 stage *drIs4* animals ($P_0$) were mutagenized in 0.6 mM N-ethyl-N-nitrosourea (ENU) diluted in M9 for 4 hours at 20˚C. One day after ENU mutagenesis, F1 mutagenized eggs were isolated by hypochlorite solution and hatched on NGM plates overnight. Starved ENU mutagenized $F_1$ *drIs4* L1 animals were washed twice in 1 x M9 and seeded onto 3–16 10 cm OP50 NGM plates. F2 synchronized larvae were obtained via hypochlorite synchronization and seeded onto OP50 NGM plates. Day one adult $F_2$ *drIs4* animals were transferred to 250 mM NaCl OP50 NGM plates for 18 hours. As controls, unmutagenized *drIs4* day 1 adults were also transferred to 50 mM NaCl and 250 mM NaCl OP50 NGM plates for 18 hours. After 18 hours, RFP and GFP fluorescence intensity, time of flight (TOF), and extinction (EXT) were acquired for each animal using a COPAS Biosort (Union Biometrica, Holliston, MA). Using the unmutagenized 50 mM NaCl NGM data as a reference, gate and sort regions for animals exposed to 250 mM NaCl were defined that isolated rare mutant animals with GFP and RFP levels similar to the population of unmutagenized *drIs4* animals on 50 mM NaCl. These mutants were termed *nio* mutants (no induction of osmolyte biosynthesis gene expression). Individual *nio* mutant hermaphrodites were selfed and their $F_3$ and $F_4$ progeny re-tested to confirm the Nio phenotype.

**Backcrossing and single gene recessive determination.**   Each *nio* mutant was backcrossed to *drIs4* males three times. F1 progeny from these backcrosses were tested on 250 mM NaCl for 18 hours as day 1 adults. As expected for a recessive mutant, 100% of the crossed progeny were WT (non-*nio*). F1 heterozygous hermaphrodites from these crosses were selfed and their progeny (F2) were tested on 250 mM NaCl for 18 hours as day 1 adults. As expected for a single gene recessive mutation, ~25% of progeny exhibited the Nio phenotype (S1 Table).

**Complementation testing.**   *nio*/+ males were crossed with hermaphrodites homozygous for the mutation being complementation tested. The F1 progeny from this cross were put on 250 mM NaCl OP50 NGM plates for 18 hours and screened for complementation. Crosses in which ~50% of these F1 progeny were WT failed to complement (i.e. were alleles of the same gene). Crosses in which 100% of these F1 progeny were WT complemented (i.e. represented

alleles of different genes). Each mutant was complementation tested to every other mutant twice–as both a hermaphrodite and as a male.

**Whole genome sequencing.** DNA was isolated from starved OP50 NGM plates with WT (*drIs4*) or mutant animals using the Qiagen Gentra Puregene Tissue Kit (Cat No 158667). The supplementary protocol for "Purification of archive-quality DNA from nematode suspensions using the Gentra Puregene Tissue Kit" available from Qiagen was used to isolate DNA. DNA samples were sequenced by BGI Americas (Cambridge, MA) with 20X coverage and paired-end reads using the Illumina HiSeq X Ten System.

**SNP and INDEL Identification in Mutants.** A Galaxy workflow was used to analyze the FASTQ forward and reverse reads obtained from BGI. The forward and reverse FASTQ reads from the animal of interest, *C. elegans* reference genome Fasta file (ce11m.fa), and SnpEff download gene annotation file (SnpEff4.3 WBcel235.86) were input into the Galaxy workflow. The forward and reverse FASTQ reads were mapped to the reference genome Fasta files with the Burrows-Wheeler Aligner (BWA) for Illumina. The resultant Sequence Alignment Map (SAM) dataset was filtered using bitwise flag and converted to the Binary Alignment Map (BAM) format [51]. Read groups were added or replaced in the BAM file to ensure proper sequence analysis by downstream tools. To identify areas where the sequenced genome varied from the reference genome, the Genome Analysis Toolkit (GATK) Unified Genotyper was used. The types of variants identified with GATK were Single Nucleotide Polymorphisms (SNPs) and Insertion and Deletions (INDELs). The SnpEff4.3 WBcel235.86 gene annotation file was used to annotate the non-synonymous SNPs and INDELs that were identified as variants by GATK. The final list of all variants with annotated non-synonymous variants was exported as a Microsoft Excel table. To identify mutations in the sequenced mutants that were not in the parent strain (*drIs4)*, the MATCH and VLOOKUP functions in Microsoft Excel were used.

**RNAi methods.** Gravid adult animals on RNAi plates (NGM + 1mM IPTG + 25ug/ml carbenicillin) were hypochlorite treated. Synchronized L1s from the hypochlorite treatment were allowed to develop on RNAi plates until day one adult. Day 1 adults were seeded onto either 50 mM or 250 mM RNAi or OP50 NaCl plates. For the developmental timed RNAi experiment (Fig 2F), hypochlorite synchronized L1 animals were seeded onto *empty vector (RNAi) (ev(RNAi))* or *ogt-1(RNAi)*. At the indicated stages, animals were manually transferred from *ev(RNAi)* to *ogt-1(RNAi)*. For the adult-specific RNAi, day 1 adult animals were transferred from *ev(RNAi)* to *ogt-1(RNAi)* plates containing either 50 mM NaCl or 250 mM NaCl. The identity of all RNAi clones was confirmed by sequencing.

## COPAS biosort acquisition and analysis

Day one adults from a synchronized egg lay or hypochlorite preparation were seeded on 50 or 250 mM NaCl OP50 or the indicated RNAi NGM plates. After 18 hours, the GFP and RFP fluorescence intensity, time of flight (TOF), and extinction (EXT) of each animal was acquired with the COPAS Biosort. Events in which the RFP intensity of adult animals (TOF 400–1200) was <20 (dead worms or other objects) were excluded from the analysis. The GFP fluorescence intensity of each animal was normalized to its RFP fluorescence intensity or TOF. To determine the fold induction of GFP for each animal, each GFP/RFP or GFP/TOF was divided by the average GFP/RFP or GFP/TOF of that strain exposed to 50 mM NaCl. The relative fold induction was determined by setting the fold induction of *drIs4* exposed to 250 mM NaCl to 1. For the data in Fig 5, the 'Degree of Rescue' was calculated as $(A_{fc} - ogt\text{-}1(dr20)_{fc\text{-}mean}) / (WT_{fc\text{-}mean} - ogt\text{-}1(dr20)_{fc\text{-}mean})$ where A = the strain of interest, 'fc' = the fold change of GFP:RFP on 250 mM NaCl versus 50 mM NaCl, and 'fc-mean' is the mean fold change of that genotype.

The raw GFP:RFP fold change data used to calculate the 'Degree of Rescue' are shown in S5A and S5B Fig. Each graphed point represents the quantified signal from a single animal.

## Molecular biology and transgenics

**Reporter strains.** The *drIs4* strain was made by injecting wild type animals with *gpdh-1p*::*GFP* (20ng/μL) and *col-12p*::*dsRed2* (100ng/μL) to generate the extrachromosomal array *drEx73*, which was integrated using UV bombardment, followed by isolation of animals exhibiting 100% RFP fluorescence. The resulting strain was outcrossed five times to wild type to generate the homozygous integrated transgene *drIs4*. *kbIs6* was generated from a Gene Gun bombardment of *unc-119(ed3)* animals with a *gpdh-1p*::*gpdh-1*::*GFP* plasmid and an *unc-119 (+)* rescue plasmid (pMM051). The resulting strain was outcrossed five times to generate *kbIs6*. *drIs4* is integrated on LGIV. The integration site for *kbIs6* is unmapped.

**Transgene rescue.** All the primers used to generate the rescue constructs can be found in S5 Table. The genomic *ogt-1* rescue construct (used in the *drEx465* extrachromosomal array) was made by amplifying *ogt-1* with 2 kb of sequence upstream of the start codon and 1 kb of sequence downstream of the stop codon. All other rescue constructs (used in extrachromosomal arrays *drEx468 –drEx474*) were made using Gibson Assembly. The *ogt-1* promoter, *ogt-1* cDNA, and *ogt-1* 3'UTR were cloned into the *pPD61.125* vector through a four component Gibson Assembly reaction. This vector was used as the backbone for all other promoter and human OGT rescue constructs. All rescue constructs were confirmed by Sanger sequencing. Extrachromosomal array lines were made by injecting day one adult animals with the rescue construct (20 ng/μL) and *rol-6(su1006)* (100 ng/μL).

**CRISPR/Cas9 genomic editing.** CRISPR allele generation and TPR deletion was performed using the single-stranded oligodeoxynucleotide donors (ssODN) method [52]. gRNA and repair template sequences are found in S5 Table. For identification of the *dr20* allele, we performed RFLP (restriction fragment length polymorphism) analysis using the *MboI* restriction enzyme, which cuts the WT allele, but not *dr20*. For identification of the *dr86*, *dr89*, *dr90*, and *dr91* alleles, we performed RFLP analysis using the *DdeI* restriction enzyme, which cuts the mutant alleles, but not WT. To make the *gpdh-1*::*GFP* CRISPR strain, we used a previously described double stranded DNA (dsDNA) asymmetric-hybrid donor method [52]. To make the *ogt-1*::*GFP* CRISPR strain, we used a dsDNA donor method using Sp9 modified primers [53]. Homozygous CRISPR/Cas9 generated alleles were isolated by selfing heterozygotes to ensure that complex alleles were not obtained.

**mRNA isolation, cDNA synthesis, and qPCR.** Day one animals were plated on 50 mM or 250 mM NaCl OP50 NGM plates for 24 hours. Unless noted otherwise, after 24 hours, 35 animals were picked into 50 μL Trizol for mRNA isolation. RNA isolation followed a combined Trizol/RNeasy column purification method as previously described [11]. cDNA was synthesized from total RNA using the SuperScript VILO Master Mix. SYBR Green master mix, 2.5 ng input RNA, and the primers listed in S5 Table were used for each qPCR reaction. qPCR reactions were carried out using an Applied Biosystems 7300 Real Time PCR machine. *act-2* primers were used as a control for all qPCR reactions. At least three biological replicates of each qPCR reaction were performed with three technical replicates per biological replicate. qPCR data was analyzed through ΔΔCt analysis with all samples normalized to *act-2*. Data are represented as fold induction of RNA on 250 mM NaCl relative to on 50 mM NaCl.

**Western blots.** Cell lysates were prepared from hypochlorite synchronized day 1 adult animals exposed to 50 mM or 250 mM NaCl plates for 18 hours. 3–5 non-starved 10 cm plates were concentrated into a 100 μL mixture. NuPage LDS Sample Buffer (4X) and NuPAGE Sample Reducing Agent (10X) were added and the sample was frozen and thawed three times at

-80˚C and 37˚C. Prior to get loading, the sample was heated to 100˚C for 10 minutes and cleared by centrifugation at 4˚C, 12,000 x g for 15 minutes. The cleared supernatant was run on a 4–12% or 8% Bis-Tris Mini Plus gel and transferred to a nitrocellulose membrane using iBlot 2 NC Regular Stacks and the iBlot 2 Dry Blotting System. The membranes were placed on iBind cards and the iBind western device was used for the antibody incubation and blocking. The Flex Fluorescent Detection (FD) Solution Kit or the iBind Solution Kit was used to dilute the antibodies and block the membrane. The antibodies used are listed in S7 Table. The following antibody dilutions were used: 1:1000 α-GFP, 1:2000 α-ß-Actin, 1:2000 α-mouse HRP, and 1:4000 Goat α-Mouse IgG (H+L) Cross-Absorbed Secondary DyLight 800. A C-DiGit Licor Blot Scanner (LI-COR Biosciences, Lincoln, NE) or an Odyssey CLx imaging System (LI-COR Biosciences, Lincoln, NE) were used to image membranes incubated with a chemiluminescent or fluorescent secondary antibody, respectively.

## Microscopy

Worms were anesthetized (10mM levamisole) and mounted on either agar plates for low magnification stereo fluorescence microscopy or silicone greased slide chambers for high magnification wide-field microscopy. Images were collected on either a Leica MZ16FA fluorescence stereo dissecting scope with a DFC345 FX camera or a Leica DMI4000B inverted compound microscope with a Leica DFC 340x digital camera using the Leica Advanced Fluorescence software (Leica Microsystems, Wetzlar, Germany). Unless noted, images within an experiment were collected using the same exposure and zoom settings. Unless noted, images depict merged GFP and RFP channels of age matched day 1 adult animals exposed to 50 or 250 mM NaCl for 18 hours.

## Immunofluorescence

Embryos from a hypochlorite preparation were freeze-cracked on a superfrost slide, fixed with 4% paraformaldehyde, blocked with bovine serum albumin (BSA), incubated with a 1:400 dilution of α-O-GlcNAc monoclonal antibody (RL2) overnight, and incubated with 1:400 dilution of 1:400 goat α-mouse IgG, IgM (H+L) Secondary Antibody, Alexa Fluor 488 for 4–6 hours [54]. The antibodies used are listed in S7 Table. Washes with PBS or antibody buffer were carried out between each incubation step. DNA was stained with 1 μg/mL Hoechst 33258 diluted in PBS. Exposure matched Z-stacks of images were processed using the following deconvolution parameters (Leica Application Suite Advanced Fluorescence, 2.1.0 build 4316): Method–blind, Total iterations– 10, Refractive index– 1.518, Resized to 16 bit depth. Images were scaled to the following intensities: RL2 maximum pixel intensity = 514, Hoechst 33258 maximum pixel intensity = 1028. Final images are represented as maximum Z-stack projections.

## *C. elegans* assays

**Acute adaptation assay.** Day one adult animals were transferred to five 50 mM NaCl OP50 NGM plates and five 200 mM NaCl OP50 plates. ~25 animals were transferred to each plate (i.e. ~125 animals total per condition per genotype). After 24 hours, 20 animals from each 50 mM or 200 mM plate were transferred to 600 mM NaCl OP50 NGM plates. Animals were scored for movement after 24 hours on the 600 mM NaCl OP50 NGM plates. The experimenter was blinded to genotype. To be counted as moving, the animal had to move greater than half a body length. Animals that were not moving were lightly tapped on the nose to confirm that they were paralyzed or dead.

**Chronic adaptation assay.** 5 L4 animals were transferred to 50 or 250 mM NaCl OP50 NGM plates. Plates were monitored over several days. For the brood and development assays,

a single L4 animal was transferred to a 50 or 250 mM NaCl OP50 NGM plate. Embryos counts and transfer of the mother to a new plate were done daily until the mother stopped laying eggs. Progeny from each animal were allowed to develop and the number of L4s was counted. Percent of developed embryos was calculated by dividing the number of L4s on a plate by the number of embryos originally laid on that plate.

**Survival assays and osmotic stress resistance (Osr) assays.** Survival and Osr assays were performed as previously described [11]. Briefly, for the survival assays, day 1 adults (24 hours post-L4) were placed on OP50 spotted NGM plates containing indicated concentrations of NaCl. The survival of each animal was determined after 24 hours at 20˚C. Animals that failed to respond to prodding with a platinum wire were scored as dead. For the OSR assay, animals were transferred from standard 50 mM NaCl OP50 spotted NGM plates to either 500 mM NaCl or 700 mM NaCl NGM plates without OP50. The percentage of animals moving after 10 minutes was determined by prodding with a platinum wire. Animals that failed to respond were scored as paralyzed.

## Statistical analysis

Comparisons of means were analyzed with either a two-tailed Students t-test (2 groups) or ANOVA (3 or more groups) using the Dunnett's or Tukey's post-test analysis as indicated in GraphPad Prism 7 (GraphPad Software, Inc., La Jolla, CA). COPAS biosort data is nonparametric and was therefore analyzed using a Mann-Whitney test (2 groups) or Kruskal-Wallis test (3 or more groups) using the Dunn's post-test analysis in GraphPad Prism 7 (GraphPad Software, Inc., La Jolla, CA). p-values of $<0.05$ were considered significant. Data are expressed as mean ± S.D. with individual points shown. Underlying numerical data for all data are found in S9–S45 Tables.

## Supporting information

**S1 Fig. *ogt-1* is required for upregulation of the transcriptional *gpdh-1::GFP* reporter (*drIs4*) by hypertonic stress.** (A) Immunoblot of GFP and $\beta$-actin in lysates from WT and *ogt-1* mutant animals expressing *drIs4* exposed to 50 or 250 mM NaCl NGM plates for 18 hours. (B) COPAS Biosort quantification of GFP and RFP signal in day 2 adult WT animals expressing *drIs4* exposed to 50 or 250 mM NaCl NGM plates for 18 hours. Animals were grown on *ev(RNAi)* or *ogt-1(RNAi)* plates for multiple generations. Data are represented as fold induction of normalized GFP/RFP ratio on 250 mM NaCl RNAi plates relative to on 50 mM NaCl RNAi plates. Each point represents the quantified signal from a single animal. Data are expressed as mean ± S.D. ****—p<0.0001 (Mann-Whitney test). N ≥ 334 for each group. *Inset*: Wide-field fluorescence microscopy of day 2 adult animals expressing *drIs4* exposed to 250 mM NaCl NGM plates for 18 hours. Animals were grown on *ev(RNAi)* or *ogt-1(RNAi)* plates for multiple generations. Images depict merged GFP and RFP channels. Scale bar = 100 microns. (C) COPAS Biosort quantification of GFP and RFP signal in day 2 adult animals expressing *drIs4* exposed to 50 or 250 mM NaCl NGM plates for 18 hours in the indicated genetic background. *drEx465* is an extrachromosomal array expressing a 10.3 Kb *ogt-1* genomic DNA fragment containing ~2Kb upstream and ~1Kb downstream of the *ogt-1* coding sequence. Data are represented as fold induction of normalized GFP/RFP ratio on 250 mM NaCl NGM plates relative to on 50 mM NaCl NGM plates. Each point represents the quantified signal from a single animal. Data are expressed as mean ± S.D. ****—p<0.0001, *—p<0.05 (Kruskal-Wallis test with post hoc Dunn's test). N ≥ 37 for each group.
(TIF)

**S2 Fig. *ogt-1* is not required for upregulation of transcriptional reporters by heat shock or ER stress or for the upregulation of the *nlp-29p::GFP* reporter by hypertonic stress.** (A) Wide-field fluorescence microscopy of day 2 adult animals expressing *hsp-16.2p::GFP (gpIs1)* grown on *ev(RNAi)*, *ogt-1(RNAi)*, or *hsf-1(RNAi)* plates and exposed to control or heat shock conditions (35˚C for 3 hours, 18 hour recovery at 20˚C). Images depict the GFP channel, since there is not a normalizing RFP reporter in these strains. Scale bar = 100 microns. (B) COPAS Biosort quantification of GFP and TOF signal from animals in (A). Data are represented as fold induction of normalized GFP/TOF ratio of animals exposed to heat shock conditions relative to animals exposed to control conditions. Each point represents the quantified signal from a single animal. Data are expressed as mean ± S.D. ****—p<0.0001 (Kruskal-Wallis test with post hoc Dunn's test). N ≥ 158 for each group. (C) Wide-field fluorescence microscopy of day 2 adult animals expressing *hsp-4p::GFP (zcIs4)* exposed to DTT plates for 18 hours. The *ogt-1 (dr50)* allele carries the same homozygous Q600STOP mutation as the *ogt-1(dr20)* allele and was introduced using CRISPR/Cas9. Images depict the GFP channel. Scale bar = 100 microns. (D) COPAS Biosort quantification of GFP and TOF signal from animals in (C). Data are represented as fold induction of normalized GFP/TOF ratio of animals exposed to DTT plates relative to animals exposed to control plates. Each point represents the quantified signal from a single animal. Data are expressed as mean ± S.D. ****—p<0.0001, n.s = nonsignificant (Kruskal-Wallis test with post hoc Dunn's test). N ≥ 48 for each group. (E) COPAS Biosort quantification of GFP and RFP signal from day 2 animals expressing *gpdh-1p::GFP (drIs4)* or *nlp-29:: GFP (frIs7)* exposed to 250 mM NaCl for 18 and 24 hours respectively. Data are represented as fold induction of normalized GFP/RFP ratio of animals exposed to 250 mM NaCl plates relative to animals exposed to 50 mM NaCl plates. Each point represents the quantified signal from a single animal. Data are expressed as mean ± S.D. ****—p<0.0001 (Kruskal-Wallis test with post hoc Dunn's test). N ≥ 173 for each group.
(TIF)

**S3 Fig. *ogt-1* is not required for upregulation of *gpdh-1* mRNA by hypertonic stress but is required for upregulation of GPDH-1-GFP protein.** (A) qPCR of *gpdh-1* mRNA in day 2 adult animals expressing *drIs4* exposed to 50 or 250 mM NaCl NGM plates for 3 hours. Strains include WT and *ogt-1(dr20)*. Data are represented as fold induction of RNA on 250 mM NaCl relative to 50 mM NaCl. Data are expressed as mean ± S.D. ****—p<0.0001 (Student's two-tailed t-test). N = 3 biological replicates of 35 animals for each group. (B) Immunoblot of GFP and *β*-actin in lysates from WT and *ogt-1(dr34)* animals expressing the *kbIs6* translational fusion exposed to 50 or 250 mM NaCl NGM plates for 18 hours. The *ogt-1(dr34)* allele carries the same homozygous Q600STOP mutation as the *ogt-1(dr20)* allele and was introduced using CRISPR/Cas9. Numbers under the GFP bands represent GFP signal normalized to *β*-actin signal for each sample, with the WT 250 mM NaCl sample set to 1. (C) qPCR of *gpdh-1* mRNA from day 2 adult animals exposed to 50 or 250 mM NaCl NGM plates for 24 hours. The animals express a CRISPR/Cas9 edited knock-in of GFP into the endogenous *gpdh-1* gene (*gpdh-1(dr81)*). *ogt-1* carries the *dr83* allele, which is the same homozygous Q600STOP mutation as the *ogt-1 (dr20)* allele and was introduced using CRISPR/Cas9. Data are represented as fold induction of RNA on 250 mM NaCl relative to 50 mM NaCl. Data are expressed as mean ± S.D. *—p<0.05 (Student's two-tailed t-test). N ≥ 3 biological replicates of 35 animals for each group. (D) qPCR of *gfp* mRNA from day 2 animals) exposed to 50 or 250 mM NaCl NGM plates for 24 hours. The strains are the same as in (C). Data are represented as fold induction of RNA on 250 mM NaCl relative to 50 mM NaCl. Data are expressed as mean ± S.D. n.s. = nonsignificant (Student's two-tailed t-test). N ≥ 3 biological replicates of 35 animals for each group.
(TIF)

**S4 Fig.** *ogt-1* **is not required for acute hypertonic stress survival but is required for chronic physiological and genetic adaptation to hypertonic stress.** (A) Percent survival of day 2 adult animals expressing *drIs4* exposed to 100–600 mM NaCl NGM plates for 24 hours. Strains include WT, *ogt-1(dr15)*, and *ogt-1(dr20)*. Data are expressed as mean ± S.D. N = 5 replicates of 20 animals for each salt concentration. (B) Brightfield microscopy images of animals grown on 50 mM or 250 mM NaCl for 5 and 10 days respectively. Strains include WT (*drIs4*) and *ogt-1(dr20);drIs4*. Scale bar = 100 microns (C) Percent of progeny that developed into L4s. L4 animals were placed on 50 or 250 mM NaCl and the total number of eggs laid and progeny that developed into L4s were counted each day as described in the 'Methods'. Data are expressed as mean ± S.D. ****—p<0.0001, **—p<0.01 (One-way ANOVA with post hoc Tukey's test). N = 10 independent broods for each strain. (D) Percent of moving unadapted and adapted day 3 adult animals after exposure to 600 mM NaCl NGM plates for 24 hours. *ogt-1(dr20 dr36)* is a strain in which the *dr20* mutation was converted back to WT using CRISPR/Cas9 genome editing. The *gpdh-1(dr81)* allele is a CRISPR/Cas9 edited C-terminal knock-in of GFP into the endogenous *gpdh-1*. The *ogt-1(dr84)* allele is a CRISPR/Cas9 edited C-terminal knock-in of GFP into the endogenous *ogt-1*. Data are expressed as mean ± S.D. ****—p<0.0001, n.s. = nonsignificant (One-way ANOVA with post hoc Dunnett's test). N = 5 replicates of 20 animals for each strain. (E) COPAS Biosort quantification of GFP and RFP signal in day 2 adult *drIs4* animals *drIs4* exposed to 50 mM NaCl NGM plates. Data are represented as the fold induction of normalized GFP/RFP ratio on 50 mM NaCl NGM plates, with *osm-11(n1604)* set to 1. The *ogt-1(dr52)* allele carries the same homozygous Q600STOP mutation as the *ogt-1(dr20)* allele, which was introduced using CRISPR/Cas9. Each point represents the quantified signal from a single animal. Data are expressed as mean ± S.D. ****—p<0.0001 (Mann-Whitney test). N ≥ 163 for each group. *Inset*: Wide-field fluorescence microscopy of day 2 adult animals expressing *drIs4* exposed to 50 mM NaCl NGM plates. Images depict merged GFP and RFP channels. Scale bar = 100 microns.
(TIF)

**S5 Fig. Rescue of the** *ogt-1* **Nio phenotype by tissue specific** *ogt-1* **expression and effect of catalytically impaired** *ogt-1* **mutations on O-GlcNAcylation and OGT-1-GFP protein levels and localization.** (A) COPAS Biosort quantification of GFP and RFP signal in day 2 adult animals expressing *drIs4* exposed to 50 or 250 mM NaCl NGM plates for 18 hours. Data are represented as the fold induction of normalized GFP/RFP ratio on 250 mM NaCl NGM plates relative to on 50 mM NaCl NGM plates. Each point represents the quantified signal from a single animal. Data are expressed as mean ± S.D. ***—p<0.001, ****—p<0.0001, n.s. = nonsignificant (Kruskal-Wallis test with post hoc Dunn's test). N ≥ 37 for each group. These data were used to calculate the 'Degree of Rescue' in Fig 5B. (B) COPAS Biosort quantification of GFP and RFP signal in day 2 adult animals expressing *drIs4* exposed to 50 or 250 mM NaCl NGM plates for 18 hours. Data are represented as the fold induction of normalized GFP/RFP ratio on 50 and 250 mM NaCl NGM plates relative to on 50 mM NaCl NGM plates. Each point represents the quantified signal from a single animal. Data are expressed as mean ± S.D. ****—p<0.0001, ***—p <0.001, **—p<0.01 (Kruskal-Wallis test with post hoc Dunn's test). N ≥ 40 for each group. These data were used to calculate the 'Degree of Rescue' in Fig 5D. (C) Wide-field fluorescence microscopy of fixed and stained embryos. RL2 was used to stain for nuclear pore O-GlcNAc modifications and Hoechst 33258 was used to visualize the DNA. Images are exposure matched. White arrowheads indicate RL2 staining in OGT-1[H612A] embryos. Scale bar = 10 microns. (D) Wide-field fluorescence microscopy of day 1 adult animals expressing endogenously CRISPR/Cas9 GFP tagged OGT-1 exposed to 50 mM NaCl NGM plates. Scale bar = 100 microns. Images are exposure matched. *Inset*: Zoomed in images

of the boxed area. Scale bar = 10 microns.
(TIF)

**S6 Fig. The tetratricopeptide repeat (TPR) domain of OGT-1 is required for *O*-GlcNAcylation, *gpdh-1p::GFP* induction, and hypertonic adaptation but does not alter OGT-1-GFP levels or localization.** (A) Wide-field fluorescence microscopy of fixed and stained embryos. RL2 was used to stain for nuclear pore O-GlcNAc modifications and Hoechst 33258 was used to visualize the DNA. Images are exposure matched. Scale bar = 10 microns. (B) Wide-field fluorescence microscopy of day 1 adult animals expressing endogenously CRISPR/Cas9 GFP tagged OGT-1 exposed to 50 mM NaCl NGM plates. Scale bar = 100 microns. Images are exposure matched. *Inset*: Zoomed in images of the boxed area. Scale bar = 10 microns. (C) Wide-field fluorescence microscopy of day 2 adult animals expressing *drIs4* exposed to 50 or 250 mM NaCl NGM plates for 18 hours. Images depict the GFP channel only for clarity. The RFP signal was unaffected in these rescue strains (not shown). Scale bar = 100 microns. (D) COPAS Biosort quantification of GFP and RFP signal in day 2 adult animals expressing *drIs4* exposed to 50 or 250 mM NaCl NGM plates for 18 hours. Data are represented as the fold induction of normalized GFP/RFP ratio on 50 and 250 mM NaCl NGM plates relative to on 50 mM NaCl NGM plates. Each point represents the quantified signal from a single animal. Data are expressed as mean ± S.D. ****—$p<0.0001$, ***—$p<0.001$, n.s. = nonsignificant (Kruskal-Wallis test with post hoc Dunn's test). N ≥ 92 for each group. (E) Percent of moving unadapted and adapted day 3 adult animals expressing *drIs4* exposed to 600 mM NaCl NGM plates for 24 hours. Data are expressed as mean ± S.D. ****—$p<0.0001$, n.s. = nonsignificant (One-way ANOVA with post hoc Tukey's test). N = 5 replicates of 20 animals for each strain.
(TIF)

**S7 Fig. Inhibition of proteasomal or autophagic pathways does not rescue *gpdh-1p::GFP* expression during hypertonic stress in *ogt-1(dr20)* mutants.** (A) COPAS Biosort quantification of GFP and RFP signal in day 2 adult animals expressing *drIs4* exposed to 50 or 250 mM NaCl NGM plates for 18 hours. Animals were placed on *empty vector(RNAi) (ev(RNAi))* or *rpn-8 (RNAi)* plates as L1s. Data are represented as normalized fold induction of normalized GFP/RFP ratio on 250 mM NaCl RNAi plates relative to on 50 mM NaCl RNAi plates. Each point represents the quantified signal from a single animal. Data are expressed as mean ± S.D. ****—$p<0.0001$ (Kruskal-Wallis test with post hoc Dunn's test). N ≥ 14 for each group. (B) COPAS Biosort quantification of GFP and RFP signal in day 2 adult animals expressing *drIs4* exposed to 50 or 250 mM NaCl NGM plates for 18 hours. Animals were placed on *empty vector(RNAi) (ev (RNAi))* or *lgg-1(RNAi)* plates as L1s. Data are represented as normalized fold induction of normalized GFP/RFP ratio on 250 mM NaCl RNAi plates relative to on 50 mM NaCl RNAi plates. Each point represents the quantified signal from a single animal. Data are expressed as mean ± S.D. ****—$p<0.0001$ (Kruskal-Wallis test with post hoc Dunn's test). N ≥ 47 for each group.
(TIF)

**S1 Table. *ogt-1(dr15)* and *ogt-1(dr20)* genetics are consistent with recessive and single gene alleles.**
(PDF)

**S2 Table. All *ogt-1* alleles fail to complement for the Nio phenotype.**
(PDF)

**S3 Table. Backcrossing does not substantially reduce the number of SNPs and INDELS in the *ogt-1(dr20)* strain.**
(PDF)

**S4 Table. Strain used in this study.**
(PDF)

**S5 Table. DNA oligos used in this study.**
(PDF)

**S6 Table. Bacterial strains used in this study.**
(PDF)

**S7 Table. Chemicals, antibodies, peptides, and recombinant proteins.**
(PDF)

**S8 Table. Critical commercial assays.**
(PDF)

**S9 Table. Underlying numerical data for Fig 1B.**
(PDF)

**S10 Table. Underlying numerical data for Fig 1C.**
(PDF)

**S11 Table. Underlying numerical data for Fig 1E.**
(PDF)

**S12 Table. Underlying numerical data for Fig 1F.**
(PDF)

**S13 Table. Underlying numerical data for Fig 2D.**
(PDF)

**S14 Table. Underlying numerical data for Fig 2E.**
(PDF)

**S15 Table. Underlying numerical data for Fig 2F.**
(PDF)

**S16 Table. Underlying numerical data for Fig 3A.**
(PDF)

**S17 Table. Underlying numerical data for Fig 3B.**
(PDF)

**S18 Table. Underlying numerical data for Fig 3C.**
(PDF)

**S19 Table. Underlying numerical data for Fig 3D.**
(PDF)

**S20 Table. Underlying numerical data for Fig 3E.**
(PDF)

**S21 Table. Underlying numerical data for Fig 4A.**
(PDF)

**S22 Table. Underlying numerical data for Fig 4B.**
(PDF)

**S23 Table. Underlying numerical data for Fig 4C.**
(PDF)

**S24 Table. Underlying numerical data for Fig 5B.**
(PDF)

**S25 Table. Underlying numerical data for Fig 5D.**
(PDF)

**S26 Table. Underlying numerical data for Fig 5F.**
(PDF)

**S27 Table. Underlying numerical data for Fig 5G.**
(PDF)

**S28 Table. Underlying numerical data for S1B Fig.**
(PDF)

**S29 Table. Underlying numerical data for S1C Fig.**
(PDF)

**S30 Table. Underlying numerical data for S2B Fig.**
(PDF)

**S31 Table. Underlying numerical data for S2D Fig.**
(PDF)

**S32 Table. Underlying numerical data for S2E Fig.**
(PDF)

**S33 Table. Underlying numerical data for S3A Fig.**
(PDF)

**S34 Table. Underlying numerical data for S3C Fig.**
(PDF)

**S35 Table. Underlying numerical data for S3D Fig.**
(PDF)

**S36 Table. Underlying numerical data for S4A Fig.**
(PDF)

**S37 Table. Underlying numerical data for S4C Fig.**
(PDF)

**S38 Table. Underlying numerical data for S4D Fig.**
(PDF)

**S39 Table. Underlying numerical data for S4E Fig.**
(PDF)

**S40 Table. Underlying numerical data for S5A Fig.**
(PDF)

**S41 Table. Underlying numerical data for S5B Fig.**
(PDF)

**S42 Table. Underlying numerical data for S6D Fig.**
(PDF)

**S43 Table. Underlying numerical data for S6E Fig.**
(PDF)

**S44 Table. Underlying numerical data for S7A Fig.**
(PDF)

**S45 Table. Underlying numerical data for S7B Fig.**
(PDF)

## Acknowledgments

Some strains were provided by the CGC, which is funded by NIH Office of Research Infrastructure Programs (P40 OD010440). We thank the labs of Arjumand Ghazi, Judy Yanowitz, and Aditi Gurkar (University of Pittsburgh) for many helpful suggestions, the lab of Gary Ruvkun (Harvard University) for the Galaxy whole genome sequencing workflow, the lab of Oliver Hobert (Columbia University) for providing additional *ogt-1* strains, and David Raizen (University of Pennsylvania) for critical reading of the manuscript.

## Author Contributions

**Conceptualization:** Sarel J. Urso, John A. Hanover, Todd Lamitina.

**Data curation:** Sarel J. Urso, Todd Lamitina.

**Formal analysis:** Sarel J. Urso, Todd Lamitina.

**Funding acquisition:** Todd Lamitina.

**Investigation:** Sarel J. Urso, Marcella Comly, Todd Lamitina.

**Methodology:** Sarel J. Urso, Todd Lamitina.

**Project administration:** Todd Lamitina.

**Resources:** Marcella Comly, John A. Hanover, Todd Lamitina.

**Supervision:** Todd Lamitina.

**Validation:** Sarel J. Urso, Todd Lamitina.

**Visualization:** Sarel J. Urso, Todd Lamitina.

**Writing – original draft:** Sarel J. Urso, Todd Lamitina.

**Writing – review & editing:** Sarel J. Urso, Todd Lamitina.

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
