## [Decision Letter · Decision Letter 0]

9 Jun 2020

Dear Dr Lamitina,

Thank you very much for submitting your Research Article entitled 'The O-GlcNAc transferase OGT is a conserved and essential regulator of the cellular and organismal response to hypertonic stress' to PLOS Genetics. Your manuscript was fully evaluated at the editorial level and by independent peer reviewers. The reviewers appreciated the attention to an important topic but identified some aspects of the manuscript that should be improved.

We therefore ask you to modify the manuscript according to the review recommendations before we can consider your manuscript for acceptance. Your revisions should address the specific points made by each reviewer.

[LINK]

Yours sincerely,

Danielle A. Garsin

Associate Editor

PLOS Genetics

Gregory P. Copenhaver

Editor-in-Chief

PLOS Genetics

Reviewer's Responses to Questions

**Comments to the Authors:**

Reviewer #1: Urso et al present in a clear and logical manner the results from a genetic screen for regulators of the osmotic stress response, specifically mutants that fail to up-regulate a gpdh-1p::GFP reporter under high salt conditions. Their results will be of interest to a relatively broad audience.

Major comments:

The authors state, “OGT-1 is required for the accumulation of GPDH-1 protein during hypertonic stress”. Formally, this has not been demonstrated as they are relying on a reporter (either just GFP, or GPDH-1::GFP). In the absence so far of direct proof, it would seem prudent to do at least one of the following:

(i) Western blot against endogenous GPDH-1

(ii) Test a gpdh-1p::dsRed reporter

Further, it is not clear what mechanism they favor to explain their observations. Is it the inducible nature of gene expression that renders any translated transcript subject to this control? In the absence of detailed molecular insights, which are obviously beyond the scope of the current report, the authors might be expected to test the effect of ogt-1 on the salt-induced expression of another protein by Western blot, or as a minimum, nlp-29, for which robust transcriptional reporters exist.

The authors point out that previous studies, “utilized global ogt-1 knockdown, which eliminates both O-GlcNAcylation-dependent and –independent functions of ogt-1. The missense alleles generated here will provide powerful tools for differentiating between these functions”. The paper would be strengthened if they could provide a first insight into this question. For example, Love et al (ref 27) reported that the expression of genes such as Y65B4BR.1 and C55A6.6 was strongly affected in ogt-1 mutants. Alternatively, Bond et al (ref 29) found that the induction of ilys-2 and clec-60 expression by S. aureus is ogt-1 dependent. What about in the new alleles?

Analyses dealing with inducible expression are often problematic. Here, the authors state, “expression of ogt-1 in intestine, muscle, or neurons did not rescue (Figure 5A - B)”, and “Overexpression of a human OGT cDNA from the native C. elegans ogt-1 promoter partially rescued gpdh-1p::GFP induction by hypertonic stress in an ogt-1 LOF mutant (Figure 5C - D)”. But the results appear to be quantitatively equivalent in the two sets of experiments, so how do they justify coming to opposite conclusions? If they consider that expression of ogt-1 in intestine has no effect, they cannot claim even partial rescue of this phenotype with human OGT. This is especially important as the authors use it to argue for an ancient osmoregulatory function for OGT independent of O-GlcNAcylation catalytic activity.

Minor point

In Figure S5, “Images are not exposure matched in order to highlight the weak RL2 staining in OGT-1H612A”. OK, but this does not allow comparison; the authors should include a set of images taken with the same high exposure for all conditions.

Reviewer #2: This manuscript from the Lamitina lab identifies the O-GcNAc transferase OGT as an essential and conserved regulator of hypertonic stress responses. Their study characterizes a novel role for OGT in an essential aspect of cell physiology. The results show a role for OGT-1 in regulating osmosensitive gene expression, unexpectedly, at a post-transcriptional level. The authors use tissue-specific promoters to demonstrate that OGT-1 function occurs in the hypodermis to regulate GDPH-1 levels, and show that expression of a catalytically dead human OGT cDNA can rescue ogt-1 LOF mutants. Taken together, these results underscore the importance of OGT function outside of its catalytic O-GlcNAcylation activity, and uncover a novel role in regulation of gene expression post-transcriptionally that is independent of O-GlcNAcylation.

The research appears to be very well-designed and executed, with an elegant combination of classic and cutting-edge genetic experiments to support its claims and includes all relevant controls. The manuscript is also very clearly written and is accessible to readers. This study should be of broad interest to researchers studying the molecular mechanisms of adaptations to stress. While some key questions are left unanswered, at least one is probably beyond the scope of this study: determining the specific mechanism by which OGT regulates osmoprotective proteins post-transcriptionally. A second question is which domains of OGT, outside of the catalytic domain, are required for its regulatory activity? Performing additional analyses of targeted ogt-1 deletion alleles should be straight-forward, and would add to the scope of the present study to better address what functions of ogt-1 are important for the hypertonic stress response. For example, is the ability of OGT to interact with one or more of its many protein partners, via the different TPRs, necessary?

There are also a few specific areas in which revisions would improve the manuscript.

Major comments:

• In the results text describing Fig. 3C, the authors should add some description of the dr34 allele, that it is the CRISPR allele analogous to the dr20 mutant allele. The description is currently only in the figure legend.

• Additional description of the results for Fig. 3A,B, D would be helpful. It is a bit confusing to compare levels of gpdh-1 mRNA with the transcriptional reporter in the dr20 allele (where they are higher than in WT), to levels with the translational reporter and the dr34 allele (where they are not different from WT).

• The authors refer to Fig. S4B to support the claim that ogt-1 worms can not reproduce under mild hypertonic stress. Fig. S4B shows images of plates after growth for 5 days on 50mM NaCl and 10 days on 250mM NaCl. There are clearly fewer worms in the ogt-1 strain, consistent with an inability to grow, but it’s not as clear that they can’t reproduce. It seems a failure in growth alone could be consistent with the data shown. Additional analysis or description of the results seem to be needed in order to conclude there are reproduction defects.

• After seeing that the catalytically dead human OGT rescues the ogt-1 LOF stress response, the authors use CRISPR to engineer two mutations analogous to the human alleles. They state “only the K957M mutation suppressed O-GlcNAcylation activity completely.”, but no figure is references to show this result. It is not clear how activity was assayed?

Minor comments

• C. elegans is singular; some sentences need to be edited to reflect this point.

Reviewer #3: The manuscript entitles “The O-GlcNac transferase OGT is a conserved and essential regulator of the cellular and organismal response to hypertonic stress” describes the discovery of ogt-1 as a required gene for the C. elegans hypertonic stress response, found through an unbiased forward genetics screen. The authors used a GFP reporter strain for their screen, containing an integrated multicopy array where the promoter of an upregulated gene in hypertonic stress, gpdh-1, is driving GFP. They hit two different alleles of ogt-1 in their screen and verify that ogt-1 is required for GFP upregulation after hypertonic stress by testing existing ogt-1 alleles, using RNAi, and CRISPR-reverting one of their alleles from the screen. Surprisingly, they found that ogt-1 mutants had no effect or even an opposite effect on hypertonic stress transcripts, even as GFP protein is reduced. This is surprising because the reporter strain used in their screen would lead to a transcript that theoretically has no relation to hypertonic stress, except for maybe the 5’-UTR, depending on how the gpdh-1 promoter was fused to the GFP gene. CRISPR-knockin of GFP into the endogenous gpdh-1 gene led to an ~50% decrease in GFP in ogt-1 mutants upon hypertonic stress compared to WT, suggesting their phenotype is not related to the multicopy array used in the screen. However, the authors should address whether the discrepancy in hypertonic stress transcripts and proteins is due to the multicopy array by testing the gpdh-1::gfp transcript levels in this knockin. The authors then go on the show that while ogt-1 is not needed for survival after hypertonic stress, it is required for adaptation to the stress, although I would ask the authors to be more clear on the “adapted” phenotype as its not obvious to a reader naïve to hypertonic stress. Also, they show that ogt-1 is required for the increased survival found in existing C. elegans mutants with “constitutive” activation of hypertonic stress genes. The authors show that ogt-1 expression in the hypodermis can rescue GFP induction hypertonic stress. And they end by showing that both human and C. elegans ogt genes can rescue their ogt-1 mutant, and that this rescue is independent of the activity of the OGlcNAcylation catalytic domain of ogt.

Overall, I would like to commend the authors on a very interesting and scientifically thorough manuscript. The authors have found a novel role for a gene essential in almost every other organism except C. elegans. And this role, being required for hypertonic stress protein expression, is conserved yet completely independent from the canonical enzymatic function of the gene. Additionally, the manuscript is clearly presented and thoroughly argued, with multiple lines of evidence for nearly every argument. The only major issue is that the authors should further address the discrepancy between hypertonic stress protein vs transcript levels in the ogt-1 mutant:

Major issues:

1. The only major issue is the data showing that ogt-1 is required for osmosensitive protein levels, but not transcript levels. While I do not think the authors need to fully explain the discrepancy, I do think it is necessary to rule out or in two obvious possibilities.

a. The first possibility is that the discrepancy is due to a secondary effect of C. elegans transgenes (multicopy arrays or 3’ UTRs). The authors should indicate which 3’-UTRs are used in all of their reporter strains/genes so that readers can see if there’s a common thread (including the constitutive RFP transgene). Also, the authors use a GFP knockin into the endogenous gpdh-1 gene to suggest that the discrepancy is not due to a transgene effect, but the decrease in GFP protein levels is low in the ogt-1 mutant and they do not show the gpdh-1::gfp transcript levels. They need further data to rule out a transgene or multicopy array effect.

b. The second possibility is that ogt-1 is required for a global increase in protein translation, independent of osmosensitive genes/transcripts. This can be tested by increasing the number of osmosensitive gene transcripts they test (via qPCR) as well as testing some endogenous protein levels, and by testing the transcripts and protein levels from control genes, not expected to increase under hypertonic stress. These types of data are required to make claims like this, “These data unexpectedly suggest that OGT-1 regulates osmosensitive gene expression at a post-transcriptional level” (pg. 8)

c. A final possibility that is not addressed is that there may be transcriptional regulation by ogt-1, but only at the tissue-specific level, such that total transcripts may be unaffected, but hypodermal transcript levels decrease. This should at least be addressed in the discussion. Although it could be easily tested with smFISH.

Minor issues:

1. The sentence in the Abstract “mutations that ablate O-GlcNAcylation activity in either human or C. elegans OGT rescue the hypertonic stress response phenotype” is unintentionally misleading as its suggesting that the mutation itself is rescuing hypertonic stress. In reality you rescue with a putative catalytically dead OGT. Please rephrase.

2. Results on pg 9. Please explain what you mean by adapted and unadapted phenotypes in the main text. This is not obvious to a reader naïve to hypertonic stress

3. In the Introduction the author write, “Hypertonicity contributes to several pathophysiological conditions and is also a feature of normal physiological states such as those that exist in the kidney and thymus (3, 4).” This sentence can use more context, especially since the meaning of the latter part of this sentence is not clear.

4. In the intro, please clarify this sentence. “These organic osmolytes track extracellular osmolarity to maintain intracellular water content and cell volume.”

5. In the intro please clarify what does “through phylogeny” mean in the statement “there is a significant chemical diversity among osmolytes through phylogeny”.

6. Please provide more context to these sentences “OGT proteolytically cleaves and activates the mammalian host cell factor C1 (HCF-1) (15, 16). OGT also has non- catalytic scaffolding functions in cell adhesion and neuronal synaptic transmission (17, 18).” What is HCF-1? What do you mean by scaffolding functions? What about the serine/threonine phosphatase activity indicated in WormBase? Given your phenotype these might be related to these functions, it would be useful to know more about them.

7. In Methods: Define RFLP.

8. Methods: Please provide a protocol or reference for “freeze-cracked”.

9. Methods: Please describe the methods used for osmotic stress resistance, given its importance to this manuscript. Also please paraphrase the survival assay. Also please note any changes between the published method and the method is this manuscript.

10. Results, pg. 7, the nested double brackets are confusing. “( Nio (no induction of osmolyte biosynthesis gene expression) mutants; Figure 2A) “

11. Methods: Please add RNAi to methods, especially concerning timing of when each animal stage (L1, L4, adult) is put on RNAi, when they were put on hypertonic plates, and when they were ran into the worm sorter.

12. Results: It is noteworthy enough to state in the results that the dr20 rescue with drEx465 overexpression leads to increased GFP expression on hypertonic media.

13. Results: This statement is not exactly true. “The function of ogt-1 in the response to hypertonic stress is specific because inhibition of ogt-1 did not affect either heat shock or endoplasmic reticulum stress inducible reporter expression (Figure S2)” RNAi of ogt-1 resulted in a significant increase in GFP expression from hsp-16.2p reporter after heat shock, suggesting ogt-1 suppresses the heat shock response.

14. For Results, pg. 9 why is gpdh-1(ok1558) a presumptive null? Reference or explain the mutation.

15. In the Discussion (pg. 14) the authors state, “OGT-1 is required for the accumulation of GPDH-1 protein during hypertonic stress, but not for the upregulation of gpdh-1 mRNA.“ This should be restated as they don’t show any data on actual GPDH-1 protein levels.

16. Figure 2E: What is the statistical significance between WT and ogt-1(dr20 dr36)? There seems to be a significant reduction in the ogt-1(dr20 dr36) compared to WT?

17. Results section, pg 9: “Importantly the requirement for ogt-1 … confirmed to be functional (Figure 3E and S4C).” Please write out in the text how it was confirmed to be functional so as to prep the reader when they go to look at the figures.

**Have all data underlying the figures and results presented in the manuscript been provided?**

Reviewer #1: None

Reviewer #2: Yes

Reviewer #3: Yes

PLOS authors have the option to publish the peer review history of their article (what does this mean?). If published, this will include your full peer review and any attached files.

Reviewer #1: No

Reviewer #2: No

Reviewer #3: No

---

## [Decision Letter · Decision Letter 1]

28 Jul 2020

Dear Dr Lamitina,

Thank you very much for submitting your Research Article entitled 'The O-GlcNAc transferase OGT is a conserved and essential regulator of the cellular and organismal response to hypertonic stress' to PLOS Genetics. Your manuscript was fully evaluated at the editorial level and by independent peer reviewers. The reviewers appreciated the attention to an important topic but identified some aspects of the manuscript that should be improved.

Specifically, reviewer #1 was not completely satisfied with your response to his/her previous concerns.

We therefore ask you to modify the manuscript according to the review recommendations before we can consider your manuscript for acceptance. Your revisions should address the specific points made by each reviewer.

[LINK]

Yours sincerely,

Danielle A. Garsin

Associate Editor

PLOS Genetics

Gregory P. Copenhaver

Editor-in-Chief

PLOS Genetics

Reviewer's Responses to Questions

**Comments to the Authors:**

Reviewer #1: I am not convinced by the authors’ minimalist answers to my previous comments. I will try to explain my concerns more clearly.

“The function of ogt-1 in the response to hypertonic stress is specific because inhibition of ogt-1 did not decrease either heat shock or endoplasmic reticulum stress inducible reporter expression (Figure S2)”.

There is a very marked increase in hsp-16.2p::GFP expression upon ogt-1(RNAi). There is no comment on this, but it clearly indicates that the influence of ogt-1 is far from being specific to the upregulation of gpdh-1 expression as stated. Here, ogt-1 is influencing the level of an exogenous protein, GFP.

I had previously commented, ““OGT-1 is required for the accumulation of GPDH-1 protein during hypertonic stress”. Formally, this has not been demonstrated as they are relying on a reporter (either just GFP, or GPDH-1::GFP)”.

The authors apparently missed my point; I had understood that the GPDH-1::GFP was produced from the CRISPR tagged allele of the endogenous gpdh-1 gene. The point is that many studies have shown that the stability of chimeric proteins can be influenced by the in vivo tolerance of the fluorescence protein to proteasomal-based degradation. So, no, they have not “measured GPDH-1 protein levels in an ogt-1 mutant”, rather the levels of a GPDH-1 translational reporter (GPDH-1::GFP), and in the absence of other evidence, they cannot claim that there is a specific effect on GPDH-1.

This is tied to a second point:

“In the absence of detailed molecular insights, which are obviously beyond the scope of the current report, the authors might be expected to test the effect of ogt-1 on the salt-induced expression of another protein by Western blot, or as a minimum, nlp-29, for which robust transcriptional reporters exist”.

The authors answered: “In an exhaustive genome-wide RNAi screen for regulators of the nlp-29p::GFP induction by infection (Zugasti et al, 2016), ogt-1 was not identified. We also showed that nlp-29 mRNA, as well as another osmotically induced mRNA (hmit-1.1) continue to be induced by high salt in ogt-1(dr20) (Fig 3A). Unfortunately, methods for detecting the NLP-29 protein are not available”.

(i) Genome-wide RNAi screens are known to suffer from a significant level of false-negatives. (ii) The screen was for regulators of induction by infection. (iii) More importantly, there was no suggestion that they needed to detect the NLP-29 protein. What was proposed was a simple experiment using a nlp-29p::gfp *transcriptional* reporter. Yes, they show that dsRed levels don’t change in the ogt-1 mutant, but this is not an adequate control since the expression of the col-12p:dsRed reporter is not affected by hypertonic stress. By testing the nlp-29p::gfp reporter (or any other GFP reporter of a hypertonic stress-induced gene) with and without hypertonic stress in wild-type and ogt-1 mutants, they will go some way to providing the essential supporting evidence that the effect of ogt-1 is likely to be linked to GPDH-1 and not GFP.

I previously commented, “If they consider that expression of ogt-1 in intestine has no effect, they cannot claim even partial rescue of this phenotype with human OGT”.

They replied saying that their statistical analysis proved their point. But the measure that they used is as much affected by the distribution of the data as the relative values, and I question the validity of the comparisons that they made. I had based my previous comment on the simple inspection of the graphs, from which I maintain that any right-minded person would not try to draw such opposite conclusions for the tissue-specific rescue and the rescue with the human protein. Indeed I suggest that the most parsimonious explanation for the results in Figure 5D is that neither the human protein, nor its catalytically dead version are capable of rescuing to any real degree the mutant phenotype. The authors have now provided data for a single set of experiments (it is not stated how many times each experiment was performed). This allows one to calculate a “degree of rescue” for sample N:

= (fold change N – fold change ogt-1 mutant)/ (fold change WT – fold change ogt-1 mutant)

So first, the effect of high salt can be expressed as a fold-change relative to the control:

For Figure 5B

dr20 = 1.95

ogt-1 promoter* = 4.41

dpy-7 promoter = 8.65

nhx-2 promoter* = 2.65

myo-3 promoter = 2.52

rab-3 promoter = 2.12

* [the average value for the 50 mM samples for ogt-1 promoter and nhx-2 promoter were 0.87 and 0.95, respectively. As there were values missing in the table, this is assumed to be a data-entry error, and the expected value of “1” was used to calculate the fold-change]

For Figure 5D

WT = 6.51

ogt-1(dr20) = 2.07

ceOGT-1 = 5.35

hsOGT = 2.84

"hsOGT H498A" = 2.87

For Fig 5B, there is no WT control, but as the values of ogt-1(dr20) are very close in the 2 experiments, one can reasonably take the WT from Figure 5D

So the “degree of rescue” is:

For Figure 5B

dr20 = 0

ogt-1 promoter = 0.54

dpy-7 promoter = 1.47

nhx-2 promoter = 0.15

myo-3 promoter = 0.12

rab-3 promoter = 0.04

For Figure 5D

WT = 1

ogt-1(dr20) = 0

ceOGT-1 = 0.74

hsOGT = 0.17

"hsOGT H498A" = 0.18

This is the numerical demonstration of what one can see from the graphs: that rescue with hsOGT is no stronger, in any meaningful way, than that with the intestinal promoter nhx-2. I consider that the authors have not adequately addressed this point in their rebuttal.

Minor comment.

If the results are valid, it might be appropriate to cite PMID: 32628333

Reviewer #2: This revised manuscript from the Lamitina lab identifies the O-GcNAc transferase OGT as an essential and conserved regulator of hypertonic stress responses. Their study characterizes a novel role for OGT in an essential aspect of cell physiology. The results show a role for OGT-1 in regulating osmosensitive gene expression, unexpectedly, at a post-transcriptional level. The authors use tissue-specific promoters to demonstrate that OGT-1 function occurs in the hypodermis to regulate GDPH-1 levels, and show that expression of a catalytically dead human OGT cDNA can rescue ogt-1 LOF mutants. Taken together, these results underscore the importance of OGT function outside of its catalytic O-GlcNAcylation activity, and uncover a novel role in regulation of gene expression post-transcriptionally that is independent of O-GlcNAcylation.

The researchers have included additional experiments to address the role of the OGT TPR domain, and added experiments to clarify the developmental phenotype of ogt-1 mutants. These experiments are a significant contribution to the manuscript. The authors have also clarified the descriptions of several results, increasing the readability. It appears they have addressed all reviewer concerns, with the exception of experiments for which there are not reagents.

The research is very well-designed and executed, with an elegant combination of classic and cutting-edge genetic experiments to support its claims and includes all relevant controls. The manuscript is also very clearly written and accessible to readers. This study should be of broad interest to researchers studying the molecular mechanisms of adaptations to stress.

Reviewer #3: Looks good to me. I have no further comments for the authors.

**Have all data underlying the figures and results presented in the manuscript been provided?**

Reviewer #1: Yes

Reviewer #2: Yes

Reviewer #3: Yes

PLOS authors have the option to publish the peer review history of their article (what does this mean?). If published, this will include your full peer review and any attached files.

Reviewer #1: No

Reviewer #2: No

Reviewer #3: No

---

## [Editor Report · Decision Letter 2]

25 Aug 2020

Dear Dr Lamitina,

We are pleased to inform you that your manuscript entitled "The O-GlcNAc transferase OGT is a conserved and essential regulator of the cellular and organismal response to hypertonic stress" has been editorially accepted for publication in PLOS Genetics. Congratulations!

Yours sincerely,

Danielle A. Garsin

Associate Editor

PLOS Genetics

Gregory P. Copenhaver

Editor-in-Chief

PLOS Genetics

Comments from the reviewers (if applicable):

**Data Deposition**

http://datadryad.org/submit?journalID=pgenetics&manu=PGENETICS-D-20-00656R2

**Press Queries**

---

## [Editor Report · Acceptance letter]

25 Sep 2020

PGENETICS-D-20-00656R2 

The *O*-GlcNAc transferase OGT is a conserved and essential regulator of the cellular and organismal response to hypertonic stress 

Dear Dr Lamitina, 

We are pleased to inform you that your manuscript entitled "The *O*-GlcNAc transferase OGT is a conserved and essential regulator of the cellular and organismal response to hypertonic stress" has been formally accepted for publication in PLOS Genetics! Your manuscript is now with our production department and you will be notified of the publication date in due course.

With kind regards,

Jason Norris

PLOS Genetics

On behalf of:
